# A methodology to constrain carbon dioxide emissions from coal-fired power plants using satellite observations of co-emitted nitrogen dioxide

Fei Liu[1,2], Bryan N. Duncan[2], Nickolay A. Krotkov[2], Lok N. Lamsal[1,2], Steffen Beirle[3], Debora Griffin[4], Chris A. McLinden[4], Daniel L. Goldberg[5], and Zifeng Lu[5]

[1]Universities Space Research Association (USRA), Goddard Earth Sciences Technology and Research (GESTAR), Columbia, MD, USA
[2]NASA Goddard Space Flight Center, Greenbelt, MD, USA
[3]Max-Planck-Institut für Chemie, Mainz, Germany
[4]Air Quality Research Division, Environment and Climate Change Canada, Toronto, ON, Canada
[5]Energy Systems Division, Argonne National Laboratory, Lemont, IL, USA

*Correspondence to*: Fei Liu (fei.liu@nasa.gov)

**Abstract.** We present a method to infer $CO_2$ emissions from individual power plants based on satellite observations of co-emitted nitrogen dioxide ($NO_2$), which could serve as complementary verification of bottom-up inventories or be used to supplement these inventories. We demonstrate its utility on eight large and isolated US power plants, where accurate stack emission estimates of both gases are available for comparison. In the first step of our methodology, we infer nitrogen oxides ($NO_x$) emissions from US power plants using Ozone Monitoring Instrument (OMI) $NO_2$ tropospheric vertical column densities (VCDs) averaged over the ozone season (May-September) and a "top-down" approach that we previously developed. Second, we determine the relationship between $NO_x$ and $CO_2$ emissions based on the direct stack emissions measurements reported by continuous emissions monitoring system (CEMS) programs, accounting for coal quality, boiler firing technology, $NO_x$ emission control device type, and any change in operating conditions. Third, we estimate $CO_2$ emissions for power plants using the OMI-estimated $NO_x$ emissions and the CEMS $NO_x/CO_2$ emission ratio. We find that the $CO_2$ emissions estimated by our satellite-based method during 2005–2017 are in reasonable agreement with the US CEMS measurements, with a relative difference of 8% $\pm 41\%$ (mean $\pm$ standard deviation). The broader implication of our methodology is that it has the potential to provide an additional constraint on $CO_2$ emissions from power plants in regions of the world without reliable emissions accounting. We explore the feasibility by comparing the derived $NO_x/CO_2$ emission ratios for the US with those from a bottom-up emission inventory for other countries and applying our methodology to a power plant in South Africa, where the satellite-based emission estimates show reasonable consistency with other independent estimates. Though our analysis is limited to a few power plants, we expect to be able to apply our method to more US (and world) power plants when multi-year data records become available from new OMI-like sensors with improved capabilities, such as the TROPOspheric Monitoring Instrument (TROPOMI) and upcoming geostationary satellites, such as the Tropospheric Emissions: Monitoring Pollution (TEMPO) instrument.

## 1 Introduction

Thermal power plants, particularly coal-fired power plants, are among the largest anthropogenic $CO_2$ emitters, contributing ~40% of energy-related $CO_2$ emissions globally in 2010 (Janssens-Maenhout et al., 2017). Coal-fired power plants are expected to be one of the primary contributors of $CO_2$ emissions in the coming decades because of abundant world coal reserves (Shindell and Faluvegi, 2010). Therefore, it is important to accurately monitor global $CO_2$ emissions from power production in order to better predict climate change (Shindell and Faluvegi, 2010) and to support the development of effective climate mitigation strategies.

CO$_2$ emissions from power plants are typically quantified based on bottom-up approaches using fuel consumption and
fuel quality, though fuel properties are not always well known, resulting in uncertainties in the estimated CO$_2$ emissions for
individual plants (Wheeler and Ummel, 2008). Even for US power plants that are considered to have the most accurate
information on fuel usage among world nations, the difference between emissions estimated based on fuel usage and those
reported as part of continuous emissions monitoring systems (CEMS) programs is typically about 20% (Ackermann and
Sundquist, 2008). Thus, emission estimates based on independent data sources, such as satellite observations, are a desirable
complement to validate and improve the current CO$_2$ emissions inventories, especially in countries without CEMS data,
which is the case in most of the world.
Anthropogenic CO$_2$ emissions have been estimated from space-based CO$_2$ observations, but the existing satellite CO$_2$
sensors are designed to provide constraints on natural CO$_2$ sources and sinks (Basu et al., 2013; Houweling et al, 2015), and
thus their capability for monitoring anthropogenic point sources is limited (Nassar et al., 2017). Observations from sensors,
including the Scanning Imaging Absorption Spectrometer for Atmospheric Chartography (SCIAMACHY; Burrows et al.,
1995), Greenhouse gases Observing SATellite (GOSAT; Yokota et al., 2009), and Orbiting Carbon Observatory-2 (OCO-2;
Crisp et al., 2015), show statistically significant enhancements over metropolitan regions (Kort et al., 2012; Schneising et al.,
2013; Janardanan et al., 2016; Buchwitz et al., 2018; Reuter et al., 2019; Wang et al., 2018). However, very few studies have
focused on individual point sources. Bovensmann et al. (2010) and Velazco et al. (2011) presented a promising satellite
remote sensing concept to infer CO$_2$ emissions for power plants based on the atmospheric CO$_2$ column distribution. Nassar et
al. (2017) presented the first quantification of CO$_2$ emissions from individual power plants using OCO-2 observations.
However, because of the narrow swath (~10 km at nadir) and 16-day repeat cycle of the OCO-2 sensor, the number of clear-
day overpasses is too small to allow for the development of a global CO$_2$ emissions database.
In contrast to CO$_2$, inferring NO$_x$ emissions from individual power plants using satellite NO$_2$ column retrievals has been
done with a higher degree of confidence (e.g., Duncan et al., 2013; de Foy et al., 2015). The Dutch-Finnish Ozone
Monitoring Instrument (OMI) on NASA's Earth Observing System Aura spacecraft (Schoeberl et al., 2006) provides near
daily, global NO$_2$ tropospheric VCDs at a spatial resolution of 13×24 km$^2$ (at nadir) (Levelt et al., 2006; 2018; Krotkov et al.,
2017), which allows emission signals from individual power plants to be resolved. Beirle et al. (2011) first analyzed isolated
large sources (i.e., megacities and the US Four Corners power plant) by averaging OMI NO$_2$ tropospheric VCDs separately
for different wind directions, which allows for the estimation of NO$_x$ emissions and lifetimes by fitting an exponentially
modified Gaussian function. Several follow-up studies (e.g., de Foy et al., 2015; Lu et al., 2015 and Goldberg et al., 2019a)
further developed this approach and inferred NO$_x$ emissions from isolated power plants and cities. More recently, we
advanced this approach for sources located in polluted areas to infer NO$_x$ emissions for 17 power plants and 53 cities across
China and the US (Liu et al., 2016; 2017).
Since NO$_x$ is co-emitted with CO$_2$, NO$_x$ emissions inferred from satellite data may be used to estimate CO$_2$ emissions
from thermal power plants. Previous analyses estimated regional CO$_2$ emissions based on satellite-derived NO$_x$ emissions
and the NO$_x$ to CO$_2$ emission ratios from bottom-up emission inventories (Berezin et al., 2013; Konovalov et al., 2016;
Goldberg et al., 2019b) or co-located satellite retrievals of CO$_2$ and NO$_2$ (Reuter et al., 2014). Hakkarainen et al. (2016)
confirmed the spatial correlation between CO$_2$ spatial anomalies and OMI NO$_2$ VCD enhancements at the regional scale
using satellite observations at higher resolution. Hakkarainen et al. (2019) also showed how overlapping OCO-2 CO$_2$ data
and data of NO$_2$ from the recently launched (October 2017) European Union Copernicus Sentinel 5 precursor
TROPOspheric Monitoring Instrument (TROPOMI) can be used to identify small scale anthropogenic CO$_2$ signatures.
More recently, the co-located regional enhancements of CO$_2$ observed by OCO-2 and NO$_2$ observed by TROPOMI were
analysed to infer localized CO$_2$ emissions for six hotspots including one power plant globally (Reuter et al., 2019). As
emissions plumes are significantly longer than the swath width of OCO-2 (10 km), OCO-2 sees only cross sections of

plumes, which may not be sufficient to infer emission strengths. Because power plant emissions can have substantial temporal variations (Velazco et al., 2011) and the cross-sectional $CO_2$ fluxes are valid only for OCO-2 overpass times, the cross-sectional fluxes may not adequately represent the annual or monthly averages, which are required for the development of climate mitigation strategies. In addition, the cross-sectional fluxes may not be a good approximation for emission strengths if meteorological conditions are not taken into account (Varon et al., 2018). As compared to the method proposed in this study, Reuter's method has the advantage of not requiring a priori emission information. However, there are currently no satellite instruments with a wide enough swath to allow wider application of Reuter's method.

In this study, we present a method to estimate $CO_2$ emissions from individual power plants using OMI $NO_2$ observations and auxiliary CEMS information necessary to estimate $NO_x$ to $CO_2$ emission ratios. Such estimates could serve as complementary verification of bottom-up $CO_2$ inventories or be used to supplement these inventories. For instance, Liu et al. (2018) used satellite data of $SO_2$ to identify large $SO_2$ sources that were missing from a bottom-up emissions inventory and created a merged bottom-up/top-down $SO_2$ emissions inventory. We apply our approach to US power plants, which have an exceptionally detailed CEMS database of $NO_x$ and $CO_2$ emissions, in order to validate our method. Using auxiliary CEMS information, we explore the relationship between $NO_x$ and $CO_2$ emissions for individual power plants, assessing variations in the ratio associated with coal quality, boiler firing type, $NO_x$ emission control device technology, and changes in operating conditions. Understanding the causes of these variations will allow for better informed assumptions when applying our method to power plants that have no or uncertain information on the factors that affect their emissions ratios. We discuss the uncertainties and applications of our approach, and the potential of $NO_2$ datasets from new and upcoming satellite instruments, which will improve the utility of our method for inferring $CO_2$ emissions from power plants around the world. Finally, we discuss future research directions.

**2 Method**

In this section, we present our method to infer $CO_2$ emissions ($E_{CO_2}^{Sat}$) from satellite-derived $NO_x$ emissions ($E_{NO_x}^{Sat}$) for individual coal-fired power plants using the following equation:

$$E_{CO_2,y}^{Sat} = \frac{E_{NO_x,y}^{Sat}}{ratio_{i,y}^{CEMS}}, \tag{1}$$

where $i$ represents coal type and $y$ represents the target year. We demonstrate our method on US power plants since there are accurate CEMS stack measurements of $NO_x$ and $CO_2$ emissions with which to validate $E_{CO_2}^{Sat}$. In Section 2.1, we describe how we estimate $E_{NO_x}^{Sat}$ from OMI $NO_2$ tropospheric VCD observations. In Section 2.2, we discuss how we estimate the ratio of $NO_x$ to $CO_2$ emissions ($ratio_y^{CEMS} = E_{NO_x,y}^{CEMS} / E_{CO_2,y}^{CEMS}$) from CEMS stack measurements in the US Emissions & Generation Resource Integrated Database (eGRID; USEPA, 2018). Since post-combustion $NO_x$ control systems, including selective noncatalytic reduction (SNCR) and selective catalytic reduction (SCR), change the relationship between $E_{NO_x}^{CEMS}$ and $E_{CO_2}^{CEMS}$, we present separate methods to determine $ratio_y^{CEMS}$ for power plants without and with post-combustion $NO_x$ control systems in Section 2.2.1 and Section 2.2.2, respectively. We discuss the validation of the estimated $E_{CO_2}^{Sat}$ in Section 3.

**2.1 Estimating satellite-derived $NO_x$ emissions ($E_{NO_x}^{Sat}$)**

From all US coal-fired power plants, we selected 21 power plants for estimating $E_{NO_x}^{Sat}$. We chose these plants based on the magnitude of their annual emissions (i.e., $E_{NO_x}^{CEMS} > 10$ Gg/yr in 2005) and relative isolation from other large sources to avoid "contamination" of a power plant's $NO_x$ plume. Power plants located in urban areas (i.e., within a radius of 100 km from a city center), or clustered in close proximity (i.e., 50 km) with other large industrial plants were excluded by visual inspection

using satellite imagery from Google Earth. We used the top 200 largest US cities (ranked by 2018 population as estimated by the United States Census Bureau, available at https://en.wikipedia.org/wiki/List_of_United_States_cities_by_population) to select power plants. As discussed below, we were able to estimate $E_{NO_x}^{Sat}$ for 8 of the 21 plants. The locations of the 8 plants are shown in Figure 1 and given in Table S1.

We followed the method of Liu et al. (2016; 2017) to estimate $E_{NO_x}^{Sat}$ for 2005 to 2017. In our analysis, we used OMI $NO_2$ tropospheric VCDs from the NASA OMI standard product, version 3.1 (Krotkov et al., 2017) together with meteorological wind information from the Modern-Era Retrospective Analysis for Research and Applications, version 2 (MERRA-2; Gelaro et al., 2017). We only analysed data for the ozone season (May-September), in order to exclude winter data, which have larger uncertainties and $NO_x$ lifetimes are longer. As in our previous study (Liu et al., 2017), we calculated 1-dimensional $NO_2$ "line densities", i.e. $NO_2$ per cm, as function of distance for each wind directions separately by integration of the mean $NO_2$ VCDs (i.e. $NO_2$ per $cm^2$) perpendicular to the wind direction. We then used the changes of $NO_2$ line densities under calm wind conditions (wind speed < 2 m/s below 500 m) and windy conditions (wind speed > 2 m/s) to fit the effective $NO_x$ lifetime. We then estimated the average $NO_2$ total mass integrated around a power plant on the basis of the 3-year mean VCDs, in agreement with previous studies (Fioletov et al., 2011; Lu et al., 2015). The $NO_2$ total mass was scaled by a factor of 1.32 in order to derive total $NO_x$ mass following Beirle et al. (2011). The uncertainty associated with the $NO_x$/$NO_2$ ratio has been discussed in detail in Section 3 of the supplement in Liu et al. (2016). The 3-year average $E_{NO_x}^{Sat}$ was derived from the corresponding 3-year average $NO_x$ mass divided by the average $NO_x$ lifetime of the entire study period (Liu et al., 2017). Fitting results of insufficient quality (e.g., correlation coefficient of the fitted and observed $NO_2$ distributions <0.9) were excluded from this analysis, consistent with the criteria in Section 2.2 of Liu et al. (2016). This final filtering left 18 power plants, of which 8 had valid results for all consecutive 3-year periods between 2005 and 2017. More details of the approach are documented in Liu et al. (2017). The fitted lifetimes and other fitting parameters for all power plants are given in Table S1.

We use the Rockport power plant (37.9 °N, 87.0 °W) in Indiana to demonstrate our approach. This power plant is particularly well suited for estimating $E_{NO_x}^{Sat}$, because it is a large and isolated $NO_x$ point source. Figure 2 presents the $NO_2$ VCD map around Rockport and the fitted results. Figure 3 displays $E_{NO_x}^{Sat}$ based on 3-year mean VCDs. Each 3-year period is represented by the middle year with an asterisk (e.g., 2006* denotes the period from 2005 to 2007). For comparison to $E_{NO_x}^{Sat}$, $E_{NO_x}^{CEMS}$ is from Air Markets Program Data (available at https://ampd.epa.gov/ampd/) and averaged over the period of May to September. For this particular plant, $E_{NO_x}^{Sat}$ is always higher than $E_{NO_x}^{CEMS}$ during the entire period, except the last two years. The coefficient of determination for the entire period is $R^2$=0.68. The relative differences for individual 3-year means (defined as $(E_{NO_x}^{Sat} - E_{NO_x}^{CEMS})/E_{NO_x}^{CEMS}$) range from -20% to 41%, because of the uncertainties of $E_{NO_x}^{Sat}$ as discussed in Section 3.2. Both datasets present a declining trend from 2012*. The total declines of 45% and 26% since 2012* in $E_{NO_x}^{Sat}$ and $E_{NO_x}^{CEMS}$ are attributed to the 25% decrease in net electricity generation for the plant. The average relative difference of $E_{NO_x}^{Sat}$ and $E_{NO_x}^{CEMS}$ for the 8 plants in this study is 0% ±33%, ranging from -58% to 72% for individual 3-year periods (Figure 1).

## 2.2 Estimating $NO_x$ to $CO_2$ emission ratios using CEMS data ($ratio^{CEMS}$)

We determined the relationship between $E_{NO_x}^{CEMS}$ and $E_{CO_2}^{CEMS}$ for coal-fired power plants using eGRID information about each plant's net electric generation, boiler firing technology (e.g., tangential or wall-fired boiler), $NO_x$ control device type, fossil fuel category (i.e., coal, oil, gas and other), and coal quality (i.e., bituminous, lignite, subbituminous, refined and waste coal). We used data of power plants with more than 99% of the fuel burned being coal as reported in eGRID. We analyzed the relationship between $E_{NO_x}^{CEMS}$ and $E_{CO_2}^{CEMS}$ by coal type, as emission characteristics vary widely by coal type.

eGRID includes two datasets of emissions for $NO_x$ and $CO_2$: 1) calculated from fuel consumption data and 2) observed by
stack monitoring (i.e., $E_{NO_x}^{CEMS}$ and $E_{CO_2}^{CEMS}$). Here we focus on eGRID CEMS data as $E_{NO_x}^{CEMS}$ are reported to be highly accurate
with an error of less than 5% (e.g., Glenn et al., 2003). $E_{CO_2}^{CEMS}$ may have larger uncertainties than fuel-based emissions
estimates because of uncertainties in the calculation of flue gas flow (Majanne et al., 2015). Nevertheless, we used $E_{CO_2}^{CEMS}$ to
relate $NO_x$ emissions to $CO_2$ emissions, since the primary uncertainty of $E_{NO_x}^{CEMS}$ and $E_{CO_2}^{CEMS}$ arises from the calculation of the
flue gas flow, which will cancel in $ratio^{CEMS}$.
**2.2.1 Coal-fired power plants without post-combustion $NO_x$ control systems**
We initially limited our analysis to $E_{NO_x}^{CEMS}$ and $E_{CO_2}^{CEMS}$ from coal-fired power plants without post-combustion $NO_x$ control
systems in operation in a given year (Table 1). We find that $E_{NO_x}^{CEMS}$ and $E_{CO_2}^{CEMS}$ have a strong linear relationship (Figure 4). In
Figure 4a, we compare $E_{NO_x}^{CEMS}$ and $E_{CO_2}^{CEMS}$ from power plants (using bituminous coal) by boiler firing type in 2005. We use
bituminous coal-fired plants for illustration, as bituminous coal is the most widely used coal in US power plants. We
analyzed power plants that use cyclone or cell burner boilers separately and exclude them in Figure 4 because they typically
produce higher $NO_x$ emissions than other boiler types (USEPA, 2009; available at
https://www3.epa.gov/ttn/chief/ap42/ch01/index.html). A strong linear relationship between $E_{NO_x}^{CEMS}$ and $E_{CO_2}^{CEMS}$ is evident with
excellent correlation ($R^2 = 0.93$, N = 278), regardless of boiler firing type. Similar linear relationships exist for other years
(e.g., year 2016 in Figure 4b) and other types of coal (Table 1). The slope of the regression of $E_{NO_x}^{CEMS}$ and $E_{CO_2}^{CEMS}$,
$ratio_{regressed}^{CEMS}$, is assumed by setting the intercept to zero. Table 1 shows $ratio_{regressed,i,y}^{CEMS}$ by coal type and year. In Section
3.1, $ratio_{regressed,i,y}^{CEMS}$ will be applied to approximate $ratio_{i,y}^{CEMS}$ when estimating $E_{CO_2}^{Sat}$ from $E_{NO_x}^{Sat}$ for the 8 plants (Figure 1)
for years before post-combustion control systems were in operation.
$ratio_{regressed}^{CEMS}$ for power plants using bituminous coal decreased from 2005 (Figure 4a) to 2016 (Figure 4b) by 31% on
average because of reductions in $NO_x$ emission factors associated with improvements in boiler operations, such as by
optimizing furnace design and operating conditions. The $NO_x$ emissions factors, defined as $NO_x$ emission rates per net
electricity generation (Gg/TW • h), declined by 33% from 2005 to 2016 (Figure 4c). We interpolated $ratio_{regressed}^{CEMS}$ to get
year-specific ratios by coal type for the entire study period, as eGRID data are only available for some years (i.e., 2005,
2007, 2009, 2010, 2012, 2014 and 2016).
In addition, $ratio_{regressed}^{CEMS}$ shows significant variation by coal type and year (Figure 5). $ratio_{regressed}^{CEMS}$ is 1.7, 1.3 and 0.91
Gg $NO_x$/Tg $CO_2$ for bituminous, subbituminous and lignite coal types in 2005, respectively. A reduction over time in
$ratio_{regressed}^{CEMS}$ is observed for all coal types (Figure 5). $ratio_{regressed}^{CEMS}$ displays a large decrease of 31%, 36% and 20% from
2005 to 2016 for bituminous, subbituminous, and lignite coal types, respectively.
**2.2.2 Coal-fired power plants with post-combustion $NO_x$ control systems**
Here, we describe how we estimated $ratio^{CEMS}$ for the entire study period for plants that had post-combustion $NO_x$
control systems installed at some time during our study period, 2005–2017. The estimation is based on $ratio_{regressed}^{CEMS}$
derived in Section 2.2.1 for plants without post-combustion control systems in operation. We introduce a $NO_x$ removal
efficiency parameter, $f$, to adjust $ratio_{regressed}^{CEMS}$ for years after the installation of post-combustion control systems,
$ratio^{CEMS-Estimated}$:
$$ratio_{i,y}^{CEMS-Estimated} = ratio_{regressed,i,y}^{CEMS} \times (1 - f_y) ,$$     (2)
$f$ is commonly measured for individual power plants to describe the performance of their post-combustion $NO_x$ control
systems. It is directly reported by some power plant databases, such as the China coal-fired Power plant Emissions Database

(CPED; Liu et al., 2015). For databases that do not report $f$, like eGRID used in this study, one can estimate it for an individual power plant by first estimating the unabated emissions per electricity generation, $e_{unabated}$, which is the emission factor before the flue gas enters the post-combustion control system:

$$f_y = \frac{e_{unabated,y} - e_{CEMS,y}}{e_{unabated,y}} , \tag{3}$$

where $e_{CEMS}$ denotes the actual emission factor in terms of CEMS $NO_x$ emissions per net electricity generation (Gg/TW h). $e_{unabated}$ for a given year, $e_{unabated,y}$, is estimated based on the emission per electricity generation for years prior, $p$, to the installation of the post-combustion control system, $e_{unabated,p}$:

$$e_{unabated,y} = k_y \times e_{unabated,p} , \tag{4}$$

where the scaling factor, $k_y$, is used to account for the change over time in $e_{unabated}$ associated with improvements in boiler operations discussed in Section 2.2.1. $k_y$ is calculated as the ratio of the averaged $e_{unabated}$ (i.e., the slope of the regression of $NO_x$ emissions on electricity generation) in year, $t$, to that in year, $p$.

To assess the reliability of $ratio^{CEMS-Estimated}$, we selected all power plants which had post-combustion devices installed between 2005 and 2016. Figure 6 shows a scatterplot of $ratio^{CEMS}$ (i.e., the ratio of $E_{NO_x}^{CEMS}$ to $E_{CO_2}^{CEMS}$ for individual plants) and $ratio^{CEMS-Estimated}$. We used the $NO_x$ emissions factor in 2005, $e_{unabated,2005}$, to predict the unabated emission factor in 2016, $e_{unabated,2016}$, following Equations (3) and (4) in order to quantify the removal efficiencies for 2016, $f_{2016}$. $ratio_{2016}^{CEMS-Estimated}$ is based on the estimated $f_{2016}$ and $ratio_{regressed,2016}^{CEMS}$ from Section 2.2.1. $ratio^{CEMS}$ and $ratio^{CEMS-Estimated}$ show good correlation ($R^2 = 0.64$), which increases our confidence that the estimated removal efficiencies approximate the actual efficiencies. The slight underestimation suggested by the slope of 0.85 arises from uncertainties in estimating unabated $NO_x$ emission factors ($e_{unabated,y}$) using Equation (4) and thus removal efficiencies ($f$), which is a major source of error of $E_{CO_2}^{Sat}$ for power plants that install post-combustion $NO_x$ control systems (see details in Section 3.2).

## 3 Results and Discussion

In Section 3.1, we present $E_{CO_2}^{Sat}$ for our eight selected power plants and, in Section 3.2, we discuss the uncertainties associated with $E_{CO_2}^{Sat}$. In Section 3.3, we compare the US ratios derived in this study with those from a bottom-up inventory for other regions to explore the potential of applying our method to regions outside the US. We finally apply our approach to one power plant in South Africa, which has several independent estimates for its $CO_2$ emissions as presented in the scientific literature. Table 2 shows three-year means of $E_{NO_2}^{Sat}$, $E_{NO_2}^{CEMS}$, $E_{CO_2}^{Sat}$ and $E_{CO_2}^{CEMS}$ for eight power plants (Figure 1). Table 3 lists the mean and the standard deviation of the relative differences between $E_{NO_x}^{CEMS}$ and $E_{NO_x}^{Sat}$, and $E_{CO_2}^{CEMS}$ and $E_{CO_2}^{Sat}$ for all eight power plants.

## 3.1 Satellite-derived $CO_2$ emissions ($E_{CO_2}^{Sat}$)

Figure 7a is a scatterplot of $E_{CO_2}^{Sat}$ and $E_{CO_2}^{CEMS}$ for the eight power plants (Figure 1), seven of which did not have post-combustion $NO_x$ control systems installed during the study period, 2005–2017. The comparison shows a good correlation, $R^2$, of 0.66. The average $E_{CO_2}^{CEMS}$ for all power plants is 2.0 Gg/h and the average $E_{CO_2}^{Sat}$ is 1.8 Gg/h. The relative difference for individual three-year means (defined as $(E_{CO_2}^{Sat} - E_{CO_2}^{CEMS})/E_{CO_2}^{CEMS}$) is 8% $\pm$41% (mean $\pm$ standard deviation). For example, Figure 3 shows $E_{CO_2}^{Sat}$ for the Rockport power plant, which typically has a positive bias as compared to $E_{CO_2}^{CEMS}$ because of a positive bias in $E_{NO_x}^{Sat}$.

Figure 7b presents the generally consistent time series between $E_{CO_2}^{Sat}$ and $E_{CO_2}^{CEMS}$, with their annual averages for the eight
power plants exhibiting a declining trend of 5%/yr and 3%/yr from 2006* to 2016* for $E_{CO_2}^{Sat}$ and $E_{CO_2}^{CEMS}$, respectively. The
reduction in net electricity generation is the driving force underlying the emission changes, which decreased by 37% for the
eight power plants from 2005 to 2016, as power producers shut down coal-fired units in favor of cheaper and more flexible
natural gas as well as solar and wind (USEIA, 2018). It is interesting to note that the temporal variations in $E_{CO_2}^{Sat}$ are not as
"smooth" as those in $E_{CO_2}^{CEMS}$, which results from fluctuations in $E_{NO_x}^{Sat}$. Such fluctuations are caused by uncertainties
associated with $E_{NO_x}^{Sat}$ as discussed in Section 3.2. For example, changes in VCDs do not necessarily relate linearly with $NO_x$
emissions (e.g., Figure 2 in Duncan et al., 2013) because of temporal variations in meteorology, and nonlinear $NO_x$
chemistry (Valin et al, 2013) and transport. Averaging VCDs for a long-term period (3 years in this study) helps reduce those
influences, but small fluctuations may still exist.
**3.2 Uncertainties**
We estimated the uncertainty of $E_{CO_2}^{Sat}$ based on the fit performance of $E_{NO_x}^{Sat}$ and comparison with $E_{CO_2}^{CEMS}$. The major
sources of uncertainty are (a) $E_{NO_x}^{Sat}$ (Liu et al., 2016); (b) $ratio_{regressed}^{CEMS}$; and (c) $f$. We give the estimated uncertainties of
each source for individual power plants in Table S2.
$E_{NO_x}^{Sat}$ : The uncertainty of $E_{NO_x}^{Sat}$ is quantified following the method described in Liu et al. (2017), accounting for errors
arising from the fit procedure, the $NO_x/NO_2$ ratio and OMI $NO_2$ VCD observations (Liu et al., 2016). The number of 1.32
used for scaling the $NO_2$ to $NO_x$ ratio is based on assumptions presented in section 6.5.1 of Seinfeld and Pandis (2006) for
"typical urban conditions and noontime sun". Note that conditions are quite similar in this study because of the overpass time
of OMI close to noon, the selection of cloud-free observations, the focus on the ozone season, and the focus on polluted
regions. A case study of CTM simulations shows an identical value of 1.32 for Paris in summer (Shaiganfar et al., 2017).
The simulated $NO_x/NO_2$ ratio at the OMI overpass time within the boundary layer (up to 2 km) in a chemistry–climate
model, EMAC (Jöckel et al., 2016), was $1.28 \pm 0.08$ for polluted ($NO_x > 1 \times 10^{15}$ molec $cm^{-2}$) regions for the July 1, 2005, and
$1.32 \pm 0.06$ on average for the ozone season. However, the coarse grid of EMAC ($2.8° \times 2.8°$ in latitude and longitude) may
not capture the true range of variation of the $NO_x/NO_2$ ratio. Therefore, we assumed an uncertainty of 20% arising from the
$NO_x/NO_2$ ratio, double than the standard deviation of the EMAC ratio.
Additionally, the tropospheric air mass factors (AMF) used in $NO_2$ retrievals are based on relatively coarsely-resolved
surface albedo data and a priori $NO_2$ vertical profile shapes, likely causing low-biased VCDs over strong emission sources
(e.g., Russell et al., 2011; McLinden et al., 2014; Griffin et al., 2019). The average AMF uncertainty of ~30% (see Table 2 in
Boersma et al., 2007) likely contributes to the underestimation of emissions from some power plants in this study. Both
random and systematic (bias) uncertainties in VCDs directly propagates into the uncertainty of $E_{NO_x}^{Sat}$ (see details in the
supplement of Liu et al. (2016) and Section 3.4 of Liu et al. (2017)).
The overall uncertainties of $E_{NO_x}^{Sat}$ range from 57% to 64% for all power plants in our analysis, which is comparable with
the level of differences between $E_{NO_x}^{Sat}$ and $E_{NO_x}^{CEMS}$. We expect this uncertainty to be less for new (e.g., TROPOMI) and
upcoming (e.g., NASA Tropospheric Emissions: Monitoring Pollution, TEMPO) OMI-like sensors, which have enhanced
capabilities relative to OMI. Further details are provided in Text S1 of the Supplement.
$ratio_{regressed}^{CEMS}$: For power plants without post-combustion devices, $ratio_{regressed}^{CEMS}$ derived from the regression (Figure 4a
& b) and the plant-specific CEMS measurements are within 15%, which is assumed as the uncertainty of the ratio for all
power plants.
$f$: For power plants with post-combustion devices, an additional uncertainty of 20% is applied to reflect the difference
between the predicted and the true removal efficiency as suggested by Figure 6.

We assume that their contributions to the overall uncertainty are independent. We then define the total uncertainty, expressed as a 95% confidence interval, as the sum of the root of the quadratic sum of the aforementioned contribution. The overall uncertainties of $E_{CO_2}^{Sat}$ are ~60% for all power plants in our analysis.

However, it is worth noting that this uncertainty estimate is rather conservative. The mean and the standard deviation of the relative differences between $E_{NO_x}^{CEMS}$ and $E_{NO_x}^{Sat}$, and $E_{CO_2}^{CEMS}$ and $E_{CO_2}^{Sat}$ for all eight power plants provide a good alternative measure of uncertainties (Table 3). The relative differences are rather small, which are 0% $\pm 33\%$ and 8% $\pm 41\%$ (mean $\pm$ standard deviation) for $NO_x$ and $CO_2$, respectively. We additionally calculate the geometric standard deviations (GSDs) of the difference between $E_{CO_2}^{CEMS}$ and $E_{CO_2}^{Sat}$ from 2006* to 2016* for individual power plants in Table S2. The small values of GSDs ranging from 1.07 to 1.31 further improve our confidence in the accuracy of the derived emissions in this study.

## 3.3 Application

In this section, we assess the feasibility of applying our method to infer $CO_2$ emissions ($E_{CO_2}^{Sat}$) for power plants outside the US. We first compare the $NO_x$ to $CO_2$ emission ratios derived from this study with those from a bottom-up emission database in Section 3.3.1. We then apply the US ratio to a power plant in South Africa in Section 3.3.2.

### 3.3.1 Comparison with bottom-up ratios

Figure 8 shows the $NO_x$ to $CO_2$ emission ratios for 2010 from the global power emissions database (GPED; Tong et al., 2018), which is the only publicly-available bottom-up emission database that reports both $NO_x$ and $CO_2$ emissions for individual power plants for every country. All countries with over 30 coal-fired power plants in GPED are shown in Figure 8. Not surprisingly, countries with more strict standards in place for $NO_x$ emissions from power plants (i.e., $NO_x$ emission limit value (ELV) < 200 mg/m³; hereafter referred to as "more strict countries") have smaller $NO_x$ to $CO_2$ ratios (i.e., 1.0 versus 2.5 on average) than countries with less strict standard (i.e., $NO_x$ ELV > 200 mg/m³; hereafter referred to as "less strict countries"). Additionally, the correlation coefficients are smaller for more strict countries (i.e., 0.82 on average) as compared to less strict countries (i.e., 0.96 on average), because power plants in more strict countries are more likely to have installed post-combustion $NO_x$ control systems, which likely lowered $ratio_y^{CEMS}$, similar to what occurred in the US over our analysis period (Section 2.2.2).

We further compare the 2005 US $ratio_{regressed}^{CEMS}$ in Table 1 with the GPED $NO_x$ to $CO_2$ emission ratios for less strict countries. We chose the 2005 value for comparison based on the following considerations. In 2005, the US EPA issued the Clean Air Interstate Rule (CAIR) to address the interstate transport of ozone and fine particulate matter pollution for eastern US states, which reduced $NO_x$ emissions and, thus, $NO_x$ to $CO_2$ ratios ($ratio_y^{CEMS}$). However, similar comprehensive control strategies have not been adopted in less strict countries. In this way, the 2005 values are expected to show better consistency with the $NO_x$ to $CO_2$ ratios of less strict countries than values for more recent years. Note that the GPED database does not give information on ratios by coal type. Therefore, we use $ratio_{regressed}^{CEMS}$ for bituminous coal, which is the most widely used coal type in coal-fired power plants in most countries.

The ratios for individual power plants in less strict countries tend to be larger than the US $ratio_{regressed}^{CEMS}$ for 2005, considering that power plants in those countries may not be equipped with any $NO_x$ control devices or even low-$NO_x$ burners, a technology which is widely installed in US power plants with and without post-combustion $NO_x$ control devices. Most ratios range from US 2005 $ratio_{regressed}^{CEMS}$ to 2005 $ratio_{regressed}^{CEMS}$ + standard deviation (Figure 8). It is no surprise that some less strict countries have ratios higher than this range, which also occurs for some US power plants without post-combustion emission controls (Figure 4). However, there are considerable uncertainties in the GPED database given the scarcity of reliable emissions information in less strict countries. For example, the GPED $NO_x$ and $CO_2$ emissions estimates for Turkey and Russia, which are outliers in Figure 8, are subject to more assumptions and, thus, larger uncertainties than countries with

high-quality country-specific emission data, such as China, which has a high-resolution emissions database (CPED; Liu et al., 2015), and India, which has a database developed by Argonne National Laboratory (Lu et al., 2011).

Figure 9 shows a schematic of our methodology to estimate the $NO_x$ to $CO_2$ emission ratios for power plants outside the US. We adopt different approaches for more and less strict countries. More strict countries, including Canada, European Union (EU) member states, Japan, South Korea, and, more recently, China, usually use CEMS to monitor emissions, particularly from the largest emitters. For power plants with CEMS measurements for both $NO_x$ and $CO_2$ emissions, it is straightforward to use the measured ratios. However, there is still a significant number of power plants in those countries without CEMS technology, particularly for $CO_2$ measurements. For example, EU member states do not require power plants to use CEMS for $CO_2$ reporting and the majority of plants in the EU therefore reports $CO_2$ emissions based on emission factors (Sloss, 2011). Therefore, we recommend applying our method described in Section 2.2 to infer region-specific ratios for those power plants. The US $ratio_{regressed}^{CEMS}$ could be a less accurate, but reasonable approximation when no CEMS data are available, considering those countries share similar $NO_x$ ELVs for power plants as the US. For less strict countries, we recommend using the 2005 US values by coal type when ratios from countries with similar $NO_x$ emission standard are not available. We also recommend assigning a range from 2005 $ratio_{regressed}^{CEMS}$ to 2005 $ratio_{regressed}^{CEMS}$ + standard deviation, instead of a fixed value, to the ratio for inferring $CO_2$ emissions, considering the knowledge on ratios from those regions are too few to narrow the constraint.

As demonstrated in Section 2.2, our method presented in this study provides a reasonable estimate of the ratio for power plants without post-combustion $NO_x$ control devices with only knowing coal type. Even for regions without reliable emission information, the information on coal type, particularly for large power plants, are very likely publicly-available. For power plants that install post-combustion $NO_x$ control technology, we additionally require the removal efficiency of the device to derive the ratio. The removal efficiency of post-combustion $NO_x$ control devices is usually directly reported, as the operation of such devices is very expensive and is expected to be subject to strict quality control and assurance standards. In contrast to bottom-up approaches, many details are required for calculating $NO_x$ and $CO_2$ emissions, including coal type, coal quality, boiler firing type, $NO_x$ emission control device type, and operating condition of boiler and emission control device.

**3.3.2 Application to Matimba power plant in South Africa**

We apply the methodology shown in Figure 9 to estimate $CO_2$ emissions from a South African power plant, Matimba, which is a strong isolated $NO_x$ point source (Figure 10). It is a well-studied power plant, having had its emissions estimated using several different methods as reported in the literature. We estimate $E_{NO_x}^{Sat}$ for Matimba from 2005 to 2017 based on OMI $NO_2$ observations following the approach in Section 2.1. Matimba uses subbituminous coal with a calorific value of ~ 20 MJ/kg (Makgato and Chirwa, 2017). We apply the ratio ranging from 2005 $ratio_{regressed}^{CEMS}$ to 2005 $ratio_{regressed}^{CEMS}$ + standard deviation to Matimba, following the methodology in Figure 9, considering that South Africa is a less strict country without any post-combustion $NO_x$ control devices (Pretorius et al., 2015). Our derived $E_{CO_2}^{Sat}$ is shown in Figure 11 and fluctuates over time. The growth after 2008* is most likely caused by the increased unit operating hours driven by the desire to meet fully the demand for electricity in South Africa after a period of rolling blackouts (2007–2008) (Duncan et al., 2016). The decline afterwards may be associated with the tripping of generating units at the Matimba because of overload and shortage of coal as reported by South African government news agency (available at https://www.sanews.gov.za/south-africa/eskom-alone-cannot-solve-our-energy-challenges). The increase in 2016* may be associated with a newly-built power plant, Medupi, which began limited operations in 2015. Note that the range of $E_{CO_2}^{Sat}$ (grey band) in Figure 11 represents the emissions based on a range of $NO_x$-to-$CO_2$ ratios, not the uncertainty. We calculate the uncertainty of $E_{CO_2}^{Sat}$ for Matimba following Section 3.2 with an additional uncertainty of ~50% to reflect the fact that the ratio may range from $ratio_{regressed}^{CEMS}$

to $ratio_{regressed}^{CEMS}$ + standard deviation. The overall uncertainty of $E_{CO_2}^{Sat}$ for Matimba is 81%, as shown by the error bars in Figure 11.

Figure 11 shows $E_{CO_2}^{Sat}$ derived in this study and other independent estimates reported in the literature, including two top-down (Nassar et al., 2017; Reuter et al., 2019) and three bottom-up estimates (Wheeler and Ummel, 2008; Tong et al., 2018; Oda et al., 2018). Despite the uncertainties associated with each of these methods, the $CO_2$ emissions estimates agree reasonably well, but we do not have sufficient information to understand the differences between these estimates. However, Tong et al. (2018) present in their CPED database both $CO_2$ and $NO_x$ emissions, which allows us to determine that the difference between $E_{NO_x}^{Sat}$ and the CPED bottom-up estimate contributes significantly to the difference in $CO_2$ estimates from the two methods. $E_{NO_x}^{Sat}$ for Matimba is 3.8 Mg/h for 2010*, which is 65% smaller than the estimate by Tong et al. (2018) for 2010. It is not surprising to see such differences considering the uncertainties of satellite-derived $NO_x$ emissions and bottom-up estimates for power plants without reliable CEMS measurements. For instance, $E_{NO_x}^{Sat}$ is potentially underestimated because of the bias in the OMI $NO_2$ standard product (version 3.1) associated with a low-resolution static climatology of surface Lambert-Equivalent Reflectivity (OMLER) (Kleipool et al., 2008). We perform a sensitivity analysis by using the preliminary new version of the OMI $NO_2$ product, which uses new geometry dependent Moderate Resolution Imaging Spectroradiometer (MODIS)-based surface reflectivity. The inferred $E_{NO_x}^{Sat}$ based on the new product is over 10% higher than version 3.1. The bottom-up estimates for Matimba are subject to significant uncertainties as well. For example, Tong et al. (2018) used national total fuel consumption of the power sector for South Africa as reported by the International Energy Agency to estimate fuel consumption at the plant level as detailed fuel consumption for each plant is not currently available. Additionally, they used default $NO_x$ emission factors obtained from the literature because of the absence of country-specific measurement data.

**4 Conclusions**

In our study, we investigated the feasibility of using satellite data of $NO_2$ from power plants to infer co-emitted $CO_2$ emissions, which could serve as complementary verification of bottom-up inventories or be used to supplement these inventories that are highly uncertain in many regions of the world. For example, our estimates will serve as an independent check of $CO_2$ emissions that will be inferred from satellite retrievals of future $CO_2$ sensors (Bovensmann et al., 2010). Currently, uncertainties in $CO_2$ emissions from power plants confound national and international efforts to design effective climate mitigation strategies.

We estimate $NO_2$ and $CO_2$ emissions during the "ozone season" from individual power plants from satellite observations of $NO_2$ and demonstrate its utility for US power plants, which have accurate CEMS with which to evaluate our method. We systematically identify the sources of variation, such as types of coal, boiler, and $NO_x$ emission control device, and change in operating conditions, which affect the $NO_x$ to $CO_2$ emissions ratio. Understanding the causes of these variations will allow for better informed assumptions when applying our method to power plants that have no or uncertain information on the factors that affect their emissions ratios. For example, we estimated $CO_2$ emissions from the large and isolated Matimba power plant in South Africa, finding that our emissions estimate shows reasonable agreement with other independent estimates.

We found that it is feasible to infer $CO_2$ emissions from satellite $NO_2$ observations, but limitations of the current satellite data (e.g., spatio-temporal resolution, signal-to-noise) only allow us to apply our method to eight large and isolated U.S. power plants. Looking forward, we anticipate that these limitations will diminish for the recently launched (October 2017) TROPOMI, and three upcoming (launches expected in the early 2020s) geostationary instruments (NASA TEMPO; European Space Agency and Copernicus Programme Sentinel-4; Korea Meteorological Administration Geostationary

Environment Monitoring Spectrometer, GEMS), which are designed to have superior capabilities to OMI. High resolution TROPOMI observations are capable of describing the spatio-temporal variability of $NO_2$, even in a relatively small city like Helsinki (Ialongo et al., 2019) and allow estimates of $NO_x$ emissions to be calculated for shorter timeframes (Goldberg et al., 2019c). Higher spatial and temporal resolutions will likely reduce uncertainties in estimates of $NO_x$ emissions as well as allow for the separation of more power plant plumes from nearby sources, thus increasing the number of power plants available for analysis. Therefore, future work will be to apply our method to these new datasets, especially after several years of vetted data become available. Additional future work will include applying our method to other regions of the world with reliable CEMS information, such as Europe, Canada and, more recently, China, to develop a more reliable and complete database with region-specific ratios.

**Data availability**

The OMI $NO_2$ and MERRA-2 data can be downloaded from the Goddard Earth Sciences Data and Information Services Center (GES DISC; available at https://disc.gsfc.nasa.gov/datasets). The CEMS emissions data can be downloaded from Air Markets Program Data (available at https://ampd.epa.gov/ampd/). The GPED data are available at http://www.meicmodel.org/dataset-gped.html.

**Author contribution**

Fei Liu, Bryan N. Duncan, and Nickolay A. Krotkov designed the framework. Fei Liu, Steffen Beirle, Lok N. Lamsal, Debora Griffin, Chris A. McLinden, and Daniel L. Goldberg developed the $NO_x$ emission fitting algorithm and Fei Liu carried it out. Fei Liu and Zifeng Lu analysed the $NO_x/CO_2$ emission ratio. Fei Liu and Bryan N. Duncan prepared the manuscript with contributions from all co-authors.

**Competing interests**

The authors declare that they have no conflict of interest.

**Acknowledgments**

This research has been funded by the NASA's Earth Science Division Atmospheric Composition: Modeling and Analysis Program (ACMAP) and the Aura Science team. The Dutch-Finnish-built OMI instrument is part of the NASA EOS Aura satellite payload. KNMI and the Netherlands Space Agency (NSO) manage the OMI project. We thank the US EPA for making the Emissions & Generation Resource Integrated Database (eGRID) available on line.

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

## 1   **Figures**

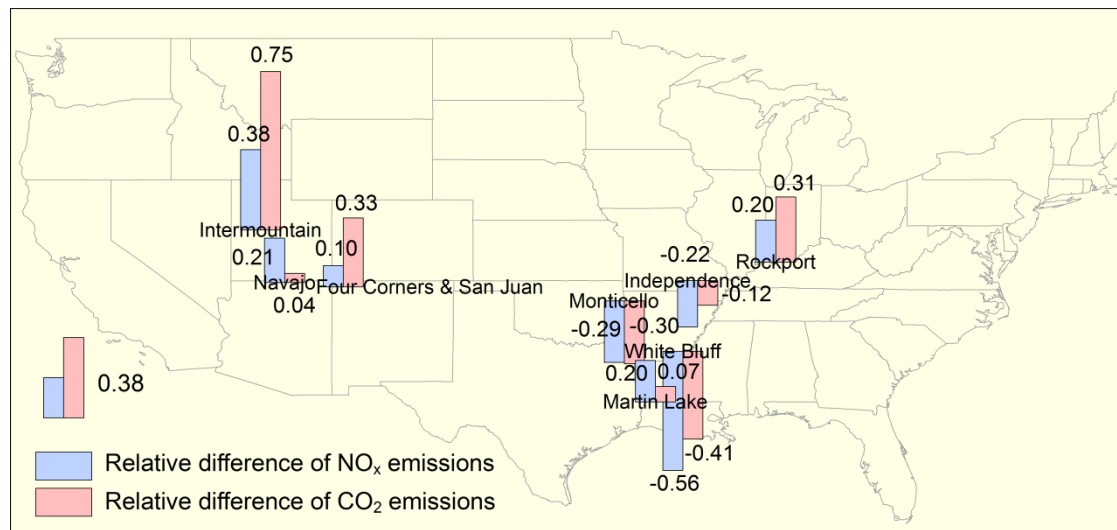

**Figure 1** Locations of the power plants investigated in this study. The bar charts denote the relative differences, defined as $(E^{Sat}-E^{CEMS})/E^{CEMS}$, averaged over 2005–2017, for $NO_x$ (blue) and $CO_2$ (red) emissions. The upward and downward bars represent positive and negative differences, respectively. The Monticello power plant installed SNCR to control $NO_x$ emissions in 2008. The other power plants are not equipped with post-combustion $NO_x$ control devices.

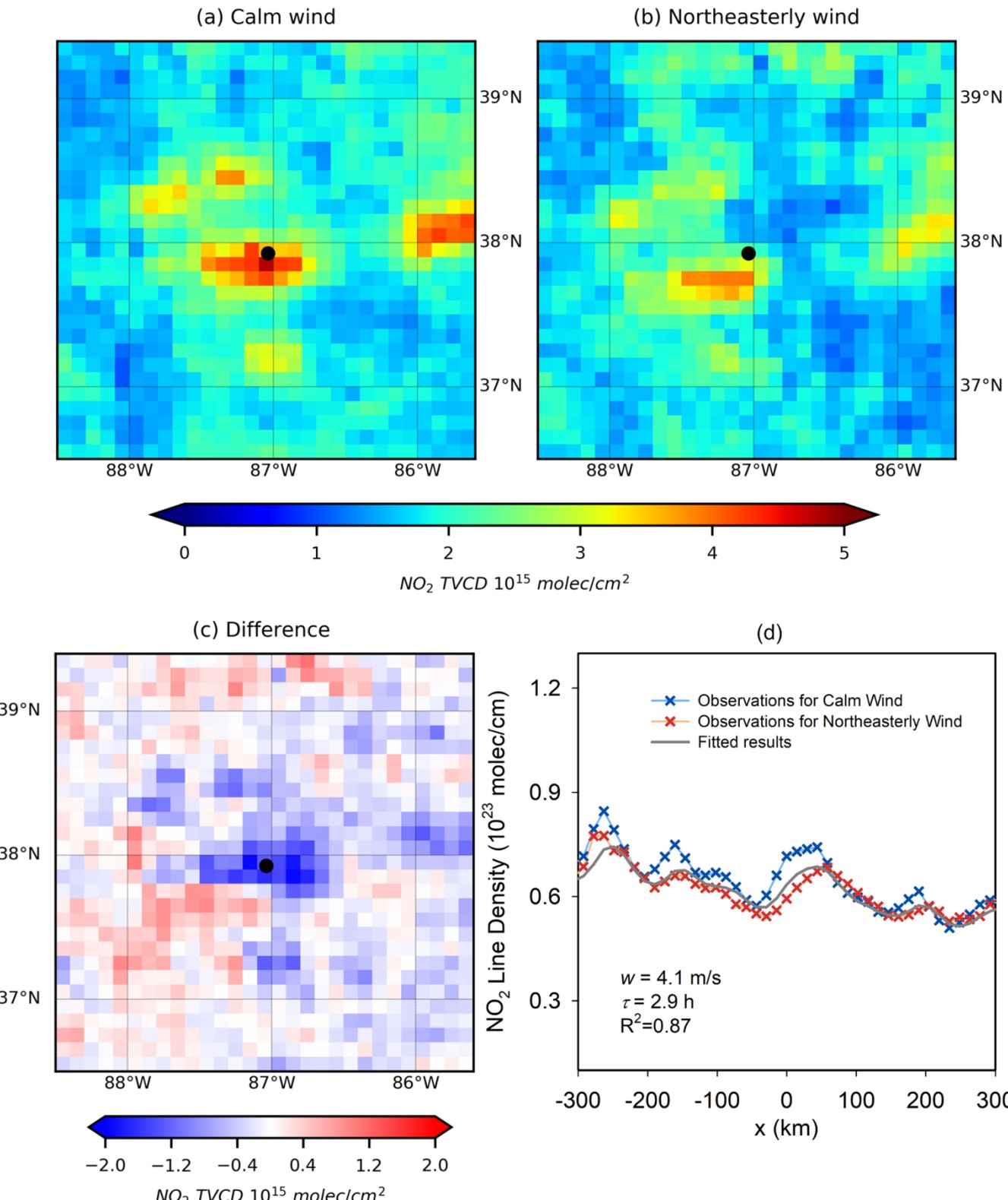

**Figure 2** Mean OMI NO$_2$ tropospheric VCDs around the Rockport power plant (Indiana, USA) for (a) calm conditions, (b) northeasterly
wind and (c) their difference (northeasterly − calm) for the period of 2005 – 2017. The location of Rockport is labelled by a black dot. (d)
NO$_2$ line densities around Rockport. Crosses: NO$_2$ line densities for calm (blue) and northeasterly winds (red) as function of the distance x
to Rockport center. Grey line: the fitted results for NO$_2$ line densities for northeasterly winds. The numbers indicate the net mean wind
velocities (windy − calm) from MERRA-2 (w), the fitted lifetime (τ), and the coefficient of determination (R$^2$) of the fit.

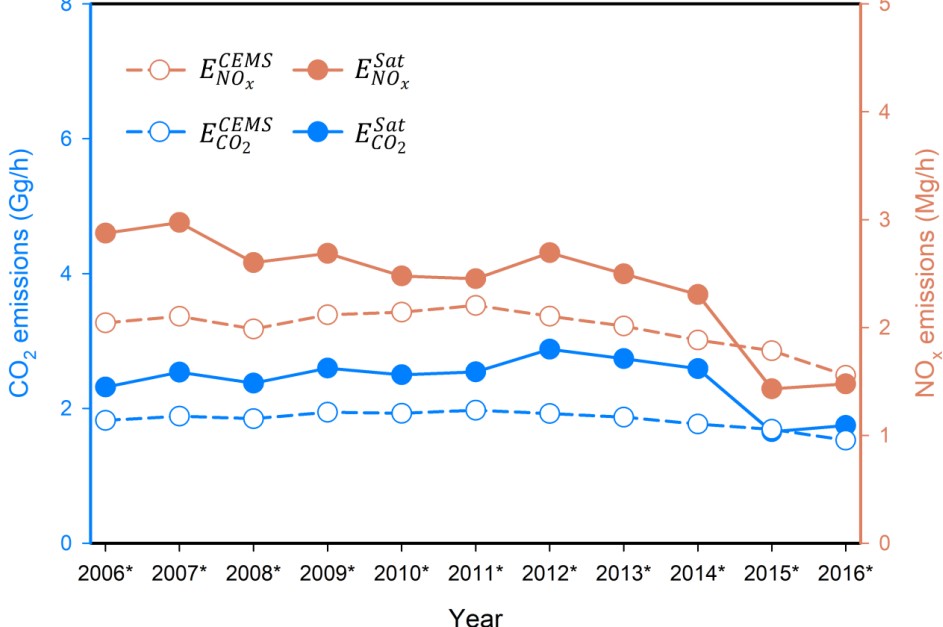

**Figure 3** $E_{NO_x}^{Sat}$ (Mg/h; orange solid lines – right axis) and $E_{CO_2}^{Sat}$ (Gg/h; blue solid line – left axis) for the Rockport power plant from 2005 to 2017. $E_{NO_x}^{CEMS}$ and $E_{CO_2}^{CEMS}$ (dashed lines) are also shown. The 3-year periods are represented by the middle year with an asterisk (e.g., 2006* denotes the period from 2005 to 2007).

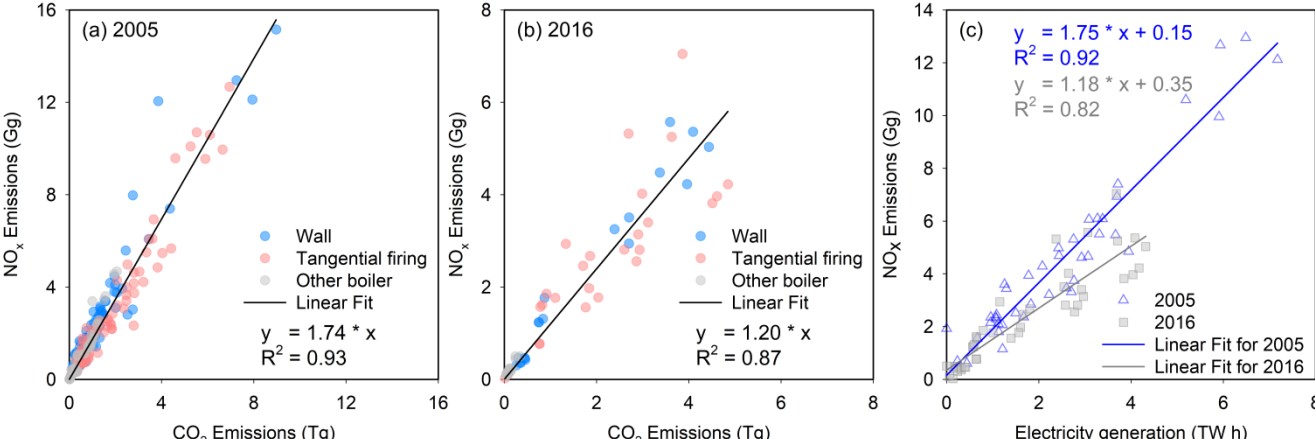

**Figure 4** Scatter plots of $E_{NO_x}^{CEMS}$ versus $E_{CO_2}^{CEMS}$ for all the US bituminous coal-fired electric generating units for (a) 2005 and (b) 2016. Values are color coded by firing type. (c) Scatter plot of $E_{NO_x}^{CEMS}$ versus electricity generation of the same units for years 2005 (triangle) and 2016 (square). Only plants without post-combustion $NO_x$ control devices within a given year are used. The electricity generation data are also from eGRID. The lines in all three panels represent the computed linear regressions.

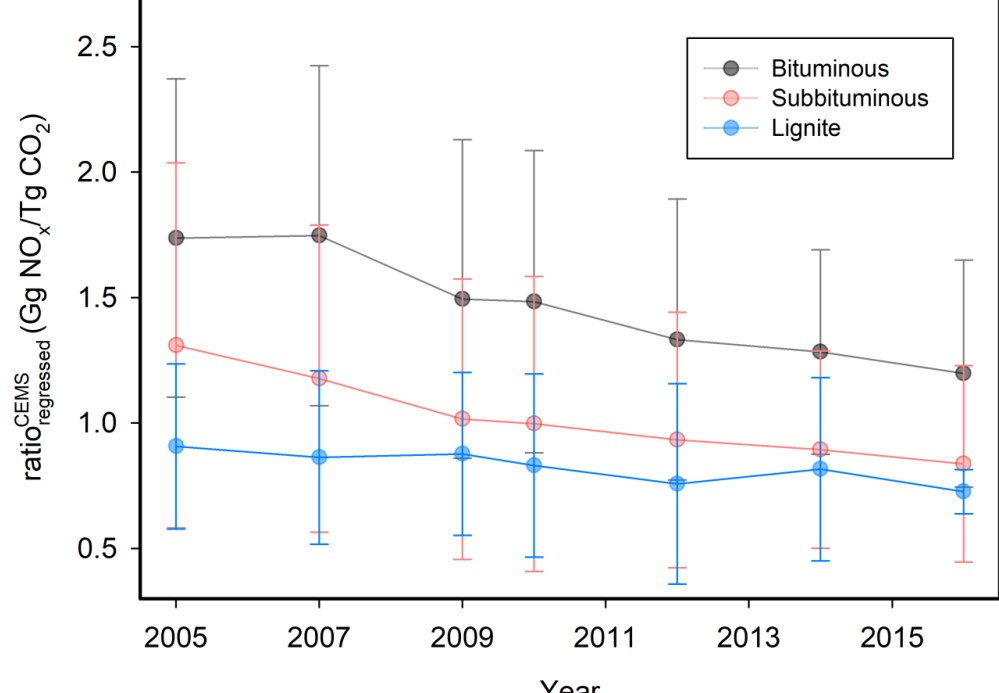

**Figure 5** Interannual trends of $ratio_{regressed}^{CEMS}$ for power plants using bituminous, subbituminous and lignite coal types and without post-
combustion $NO_x$ control devices in a given year. Error bars show the standard deviations for ratios of $E_{NO_x}^{CEMS}$ to $E_{CO_2}^{CEMS}$ for individual power
plants.

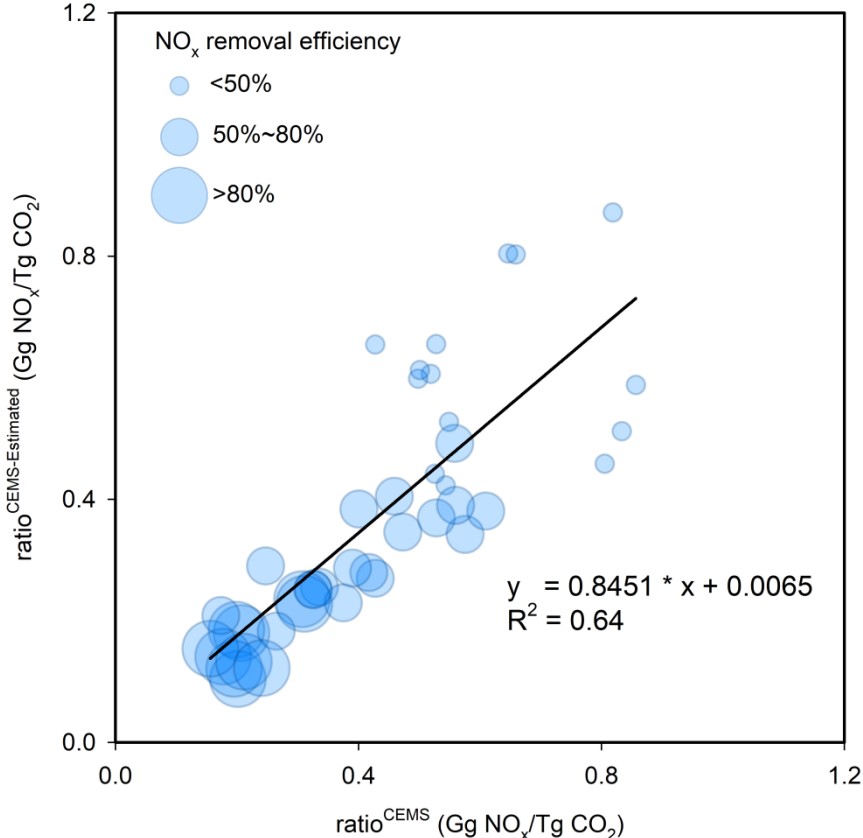

**Figure 6** Scatterplot of $ratio^{CEMS-Estimated}$ as compared to $ratio^{CEMS}$ for 2016. All 44 coal-fired power plants that operated post-
combustion devices after 2005 and before 2016 (including 2016) are used in the plot. The sizes of the circles denote the magnitude of the
$NO_x$ reduction efficiency of post-combustion control devices estimated in this study. The line represents the linear regression of $ratio^{CEMS}$
to $ratio^{CEMS-Estimated}$.

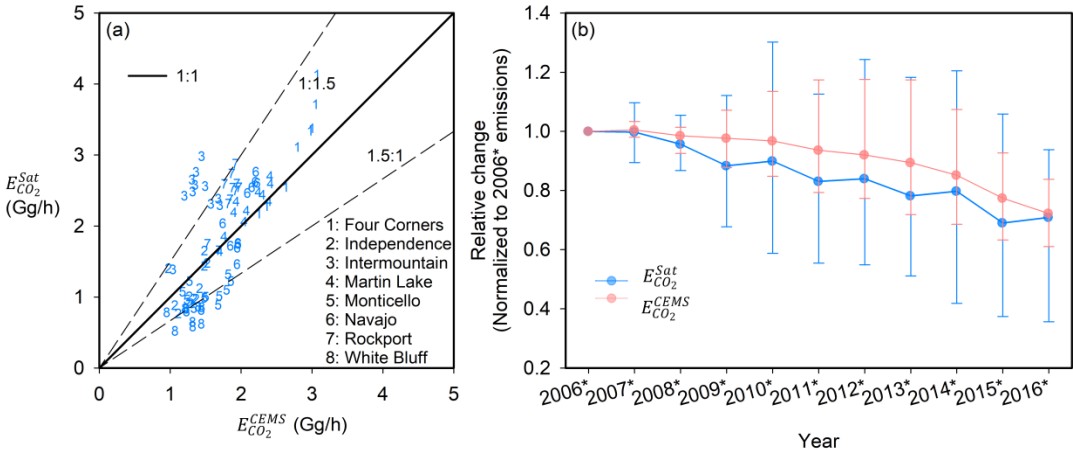

**Figure 7** (a) Scatterplot of $E_{CO_2}^{Sat}$ for eight power plants as compared to $E_{CO_2}^{CEMS}$ from 2006* to 2016*. The solid lines represent the ratio of 1:1. The dashed lines represent the ratio of 1:1.5 and 1.5:1, respectively. (b) Interannual trends of the averaged $E_{CO_2}^{Sat}$ (blue lines) and $E_{CO_2}^{CEMS}$ (pink lines) are for all power plants analyzed in this study from 2006*–2016*, as normalized to the 2006* value. The whiskers denote the maximum and minimum values.

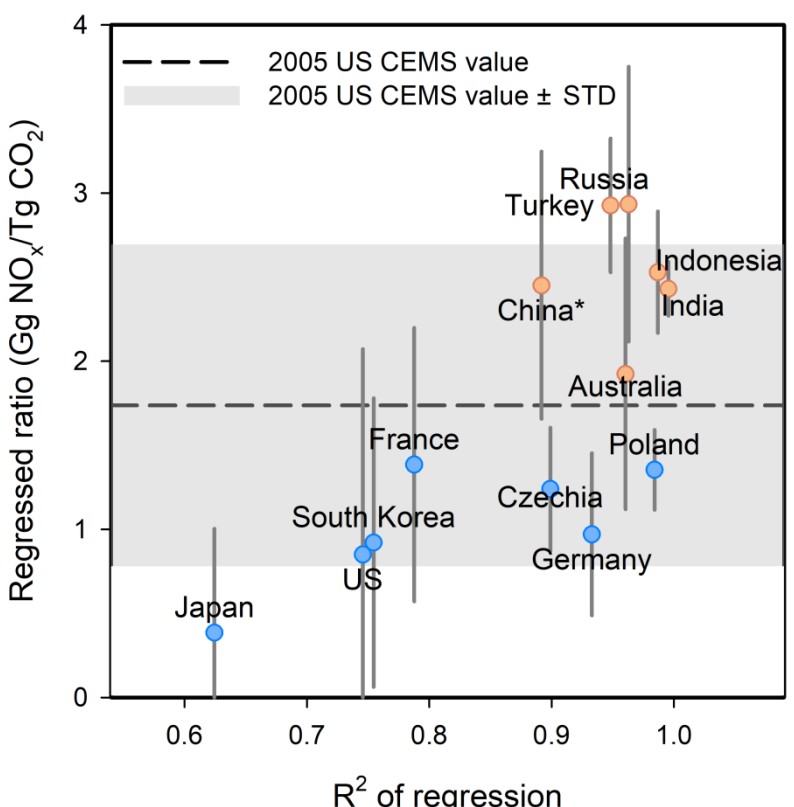

**Figure 8** Comparison of the regressed $NO_x$ to $CO_2$ emission ratios derived from the global power emissions database (GPED) for different regions versus the correlation coefficient of the regression. The blue and red circles denote regions that are subject to more strict standard for $NO_x$ emissions from power plants (i.e., a $NO_x$ ELV of 200 mg/m$^3$ or less) and other regions, respectively. Y axis: the slope of the regression of the $NO_x$ to $CO_2$ emissions with an assumed y-intercept of zero. Error bars show the standard deviations for the $NO_x$ to $CO_2$ emission ratios for individual power plants. X axis: correlation coefficient of the regression. The dashed line represents 2005 US $ratio_{regressed}^{CEMS}$ for bituminous coal derived in this study. The grey shadow represents 2005 US $ratio_{regressed}^{CEMS}$ ±standard deviation.

*China switched from being a less strict country to a more strict country in 2014, when most coal-fired power plants in China were required to comply with its new emission standards (GB13223-2011).

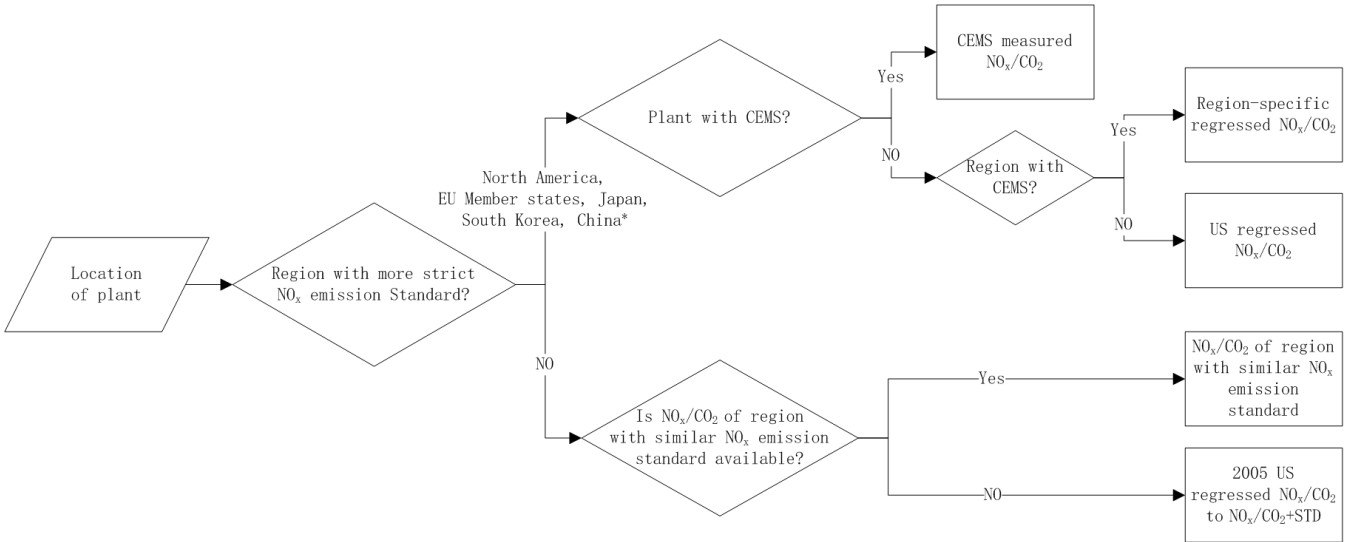

**Figure 9** Schematic of our methodology to estimate the $NO_x$ to $CO_2$ emission ratios for power plants outside the US.

*China switched from being a less strict country to a more strict country in 2014, when most coal-fired power plants in China were required to comply with its new emission standards (GB13223-2011).

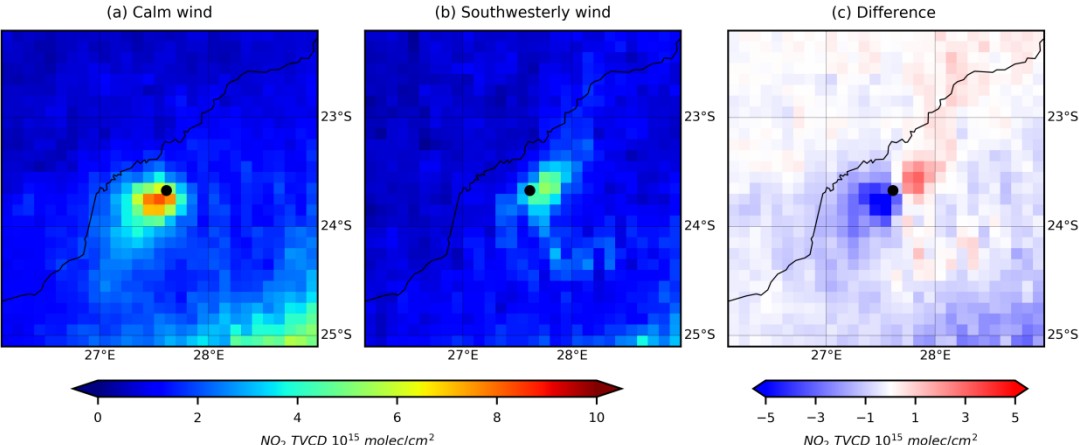

**Figure 10** Mean OMI $NO_2$ tropospheric VCDs around the Matimba power plant (Lephalale, South Africa) for (a) calm, (b) southwesterly wind conditions and (c) their difference (southwesterly − calm) for the period of 2005 – 2017. The location of Matimba is represented by a black dot.

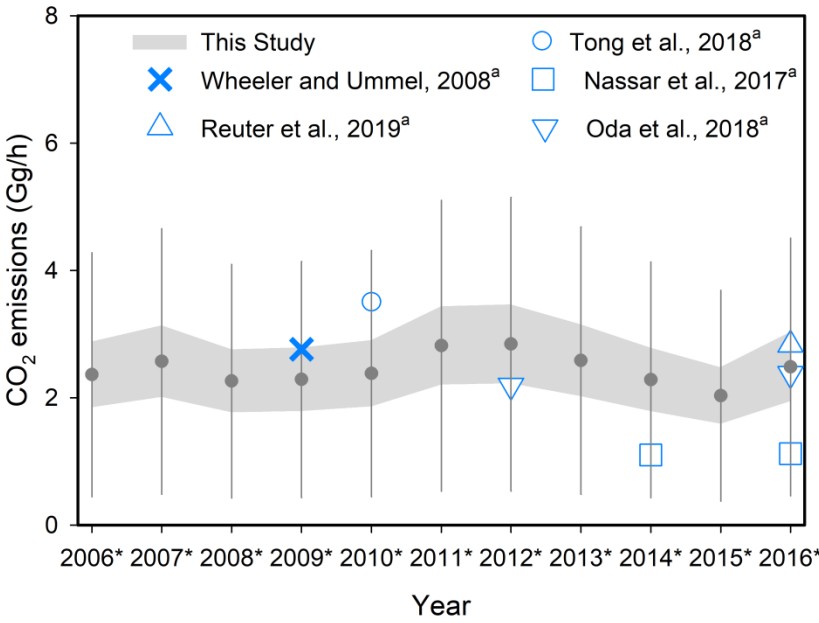

**Figure 11** Comparison of $E_{CO_2}^{Sat}$ (Gg/h) derived in this study with existing estimates for the Matimba power plant during 2005 to 2017. $E_{CO_2}^{Sat}$ is inferred based on the $NO_x$ to $CO_2$ emissions ratio ranging from $ratio_{regressed}^{CEMS}$ to $ratio_{regressed}^{CEMS}$ + standard deviation of ratio. The upper and lower grey bands denote the emissions inferred from $ratio_{regressed}^{CEMS}$ and $ratio_{regressed}^{CEMS}$ + standard deviation of ratio, respectively. The grey dots and error bars show the mean of the upper and lower grey bands and their uncertainties, respectively.

[a]Emissions are estimated for 2009 by Wheeler and Ummel (2008); for 2010 by Tong et al. (2018); for 2014 and 2016 by Nassar at al. (2017); for 2016 by Reuter et al. (2019); and for 2012 and 2016 by Oda at al. (2018).

**Table 1** The slope ($ratio_{regressed}^{CEMS}$), coefficient of determination, standard deviation and sample number of the linear regression of $E_{NO_x}^{CEMS}$ and $E_{CO_2}^{CEMS}$ by year for all US power plants without post-combustion $NO_x$ control devices from 2005 to 2016.

| Coal type | Year | $ratio_{regressed}^{CEMS}$ | $R^2$ | Standard deviation | Sample number[a] |
|---|---|---|---|---|---|
| Bituminous | 2005 | 1.74 | 0.93 | 0.63 | 278 |
| | 2007 | 1.75 | 0.91 | 0.68 | 286 |
| | 2009 | 1.49 | 0.88 | 0.64 | 241 |
| | 2010 | 1.48 | 0.86 | 0.60 | 235 |
| | 2012 | 1.33 | 0.87 | 0.56 | 190 |
| | 2014 | 1.28 | 0.87 | 0.41 | 136 |
| | 2016 | 1.20 | 0.87 | 0.45 | 66 |
| Subbituminous | 2005 | 1.31 | 0.65 | 0.73 | 226 |
| | 2007 | 1.18 | 0.61 | 0.61 | 221 |
| | 2009 | 1.02 | 0.66 | 0.56 | 230 |
| | 2010 | 1.00 | 0.67 | 0.59 | 216 |
| | 2012 | 0.93 | 0.74 | 0.51 | 200 |
| | 2014 | 0.89 | 0.74 | 0.39 | 165 |
| | 2016 | 0.84 | 0.70 | 0.39 | 111 |
| Lignite | 2005 | 0.91 | 0.74 | 0.33 | 20 |
| | 2007 | 0.86 | 0.82 | 0.35 | 22 |
| | 2009 | 0.88 | 0.91 | 0.32 | 16 |
| | 2010 | 0.83 | 0.94 | 0.37 | 18 |
| | 2012 | 0.76 | 0.91 | 0.40 | 15 |
| | 2014 | 0.82 | 0.92 | 0.37 | 12 |
| | 2016 | 0.73 | 0.78 | 0.09 | 9 |

[a]The sample number generally decreases from 2005 to 2016 as power plants installed post-combustion $NO_x$ control devices over time.

**Table 2** Summary of effective NO$_x$ lifetimes, satellite-derived NO$_x$ emissions ($E_{NO_x}^{Sat}$), CO$_2$ emissions ($E_{CO_2}^{Sat}$) and bottom-up NO$_x$ emissions ($E_{NO_x}^{CEMS}$), CO$_2$ emissions ($E_{CO_2}^{CEMS}$) for 8 US power plants during May to September from 2005 to 2017. The 3-year periods are represented by the middle year with an asterisk.

| Category | Year | Four Corners & San Juan | Independence | Intermountain | Martin Lake | Monticello | Navajo | Rockport | White Bluff |
|---|---|---|---|---|---|---|---|---|---|
| NO$_x$ lifetime | 2005-2017 | 2.7 | 2.5 | 2.2 | 2.3 | 3.2 | 2.3 | 2.4 | 4.3 |
| $E_{NO_x}^{Sat}$ (Mg/h) | 2006* | 10.5 | 2.0 | 4.0 | 2.4 | 1.1 | 4.6 | 2.9 | 1.0 |
| | 2007* | 10.0 | 1.7 | 4.1 | 2.3 | 1.1 | 4.4 | 3.0 | 0.9 |
| | 2008* | 9.4 | 1.6 | 3.7 | 2.0 | 0.8 | 4.5 | 2.6 | 0.9 |
| | 2009* | 7.2 | 1.2 | 3.9 | 2.1 | 0.7 | 3.9 | 2.7 | 0.7 |
| | 2010* | 6.8 | 1.0 | 4.4 | 2.1 | 0.6 | 3.6 | 2.5 | 0.9 |
| | 2011* | 6.5 | 0.9 | 3.6 | 1.8 | 0.7 | 2.5 | 2.5 | 0.8 |
| | 2012* | 6.3 | 0.9 | 3.4 | 1.6 | 0.6 | 2.3 | 2.7 | 0.8 |
| | 2013* | 5.6 | 0.8 | 3.5 | 1.8 | 0.5 | 1.9 | 2.5 | 0.6 |
| | 2014* | 4.4 | 0.7 | 3.5 | 1.7 | 0.8 | 2.2 | 2.3 | 0.5 |
| | 2015* | 3.8 | 0.8 | 3.0 | 1.4 | 0.7 | 2.1 | 1.4 | 0.4 |
| | 2016* | 3.5 | 1.2 | 1.7 | 1.2 | 0.6 | 2.5 | 1.5 | 0.7 |
| $E_{NO_x}^{CEMS}$ (Mg/h) | 2006* | 7.4 | 1.8 | 3.0 | 1.8 | 1.5 | 3.8 | 2.0 | 1.7 |
| | 2007* | 7.3 | 1.8 | 3.1 | 1.8 | 1.4 | 3.9 | 2.1 | 1.6 |
| | 2008* | 6.8 | 1.8 | 2.9 | 1.8 | 1.3 | 3.8 | 2.0 | 1.6 |
| | 2009* | 6.5 | 1.6 | 2.9 | 1.8 | 1.2 | 3.4 | 2.1 | 1.8 |
| | 2010* | 6.2 | 1.6 | 2.8 | 1.7 | 1.1 | 2.8 | 2.1 | 1.8 |
| | 2011* | 6.2 | 1.4 | 2.5 | 1.5 | 1.0 | 2.2 | 2.2 | 1.9 |
| | 2012* | 6.1 | 1.3 | 2.4 | 1.4 | 0.9 | 1.9 | 2.1 | 1.9 |
| | 2013* | 5.6 | 1.3 | 2.4 | 1.3 | 0.9 | 1.9 | 2.0 | 2.0 |
| | 2014* | 5.2 | 1.2 | 2.5 | 1.3 | 0.8 | 1.9 | 1.9 | 1.9 |
| | 2015* | 4.3 | 1.2 | 2.0 | 1.3 | 0.8 | 1.7 | 1.8 | 1.5 |
| | 2016* | 3.9 | 1.1 | 1.5 | 1.2 | 0.8 | 1.6 | 1.6 | 1.2 |
| $(E_{NO_x}^{Sat}-E_{NO_x}^{CEMS}) / E_{NO_x}^{CEMS}$ | 2005-2017 | 10% | -22% | 38% | 20% | -29% | 21% | 20% | -56% |
| $E_{CO_2}^{Sat}$ (Gg/h) | 2006* | 6.1 | 1.6 | 2.3 | 2.7 | 1.2 | 2.6 | 2.3 | 0.8 |
| | 2007* | 5.9 | 1.5 | 2.4 | 2.6 | 1.3 | 2.6 | 2.5 | 0.8 |
| | 2008* | 5.6 | 1.4 | 2.3 | 2.3 | 1.1 | 2.8 | 2.4 | 0.8 |
| | 2009* | 4.1 | 1.1 | 2.6 | 2.4 | 1.0 | 2.5 | 2.6 | 0.6 |
| | 2010* | 3.7 | 1.0 | 3.0 | 2.5 | 0.9 | 2.5 | 2.5 | 0.9 |
| | 2011* | 3.4 | 1.0 | 2.6 | 2.2 | 1.0 | 1.7 | 2.5 | 0.8 |
| | 2012* | 3.3 | 1.0 | 2.5 | 2.1 | 1.0 | 1.7 | 2.9 | 0.9 |
| | 2013* | 3.1 | 0.9 | 2.6 | 2.3 | 0.8 | 1.5 | 2.7 | 0.6 |
| | 2014* | 2.5 | 0.8 | 2.8 | 2.2 | 1.2 | 1.8 | 2.6 | 0.6 |
| | 2015* | 2.3 | 0.9 | 2.4 | 1.8 | 1.1 | 1.7 | 1.7 | 0.5 |
| | 2016* | 2.2 | 1.4 | 1.4 | 1.6 | 1.0 | 2.0 | 1.7 | 0.8 |
| $E_{CO_2}^{CEMS}$ (Gg/h) | 2006* | 3.1 | 1.5 | 1.7 | 2.4 | 1.9 | 2.2 | 1.8 | 1.2 |
| | 2007* | 3.1 | 1.5 | 1.7 | 2.4 | 1.8 | 2.2 | 1.9 | 1.2 |
| | 2008* | 3.0 | 1.5 | 1.6 | 2.4 | 1.8 | 2.2 | 1.8 | 1.2 |
| | 2009* | 3.1 | 1.4 | 1.5 | 2.3 | 1.7 | 2.1 | 1.9 | 1.3 |
| | 2010* | 3.0 | 1.4 | 1.4 | 2.2 | 1.7 | 2.1 | 1.9 | 1.4 |
| | 2011* | 3.0 | 1.3 | 1.3 | 2.1 | 1.5 | 2.0 | 2.0 | 1.4 |
| | 2012* | 3.0 | 1.3 | 1.3 | 2.0 | 1.5 | 1.9 | 1.9 | 1.4 |

| | | | | | | | | |
|---|---|---|---|---|---|---|---|---|
| | 2013* | 2.8 | 1.3 | 1.3 | 1.9 | 1.3 | 1.9 | 1.9 | 1.4 |
| | 2014* | 2.6 | 1.1 | 1.4 | 1.9 | 1.3 | 2.0 | 1.8 | 1.3 |
| | 2015* | 2.4 | 1.1 | 1.2 | 1.8 | 1.2 | 1.8 | 1.7 | 1.1 |
| | 2016* | 2.2 | 1.0 | 1.0 | 1.7 | 1.2 | 1.7 | 1.5 | 0.9 |
| $(E_{CO_2}^{Sat} - E_{CO_2}^{CEMS}) / E_{CO_2}^{CEMS}$ | 2005-2017 | 33% | -12% | 75% | 7% | -30% | 4% | 31% | -41% |

**Table 3** Summary of relative difference between satellite-derived $NO_x$ emissions ($E_{NO_x}^{Sat}$) and bottom-up $NO_x$ emissions ($E_{NO_x}^{CEMS}$), satellite-derived $CO_2$ emissions ($E_{CO_2}^{Sat}$) and bottom-up $CO_2$ emissions ($E_{CO_2}^{CEMS}$) for 8 US power plants during May to September from 2005 to 2017. The 3-year periods are represented by the middle year with an asterisk.

| Year | Relative Difference for $NO_x$ | | Relative Difference for $CO_2$ | |
|------|------|--------------------|------|--------------------|
|      | Mean | Standard Deviation | Mean | Standard Deviation |
| 2006* | 15% | 29% | 17% | 39% |
| 2007* | 10% | 29% | 16% | 38% |
| 2008* | 5%  | 30% | 14% | 39% |
| 2009* | -3% | 34% | 6%  | 39% |
| 2010* | -1% | 38% | 9%  | 46% |
| 2011* | -5% | 31% | 3%  | 40% |
| 2012* | -3% | 31% | 5%  | 41% |
| 2013* | -4% | 38% | 4%  | 49% |
| 2014* | -3% | 36% | 7%  | 46% |
| 2015* | -8% | 35% | 2%  | 41% |
| 2016* | -2% | 29% | 8%  | 22% |