# Peer review of "A methodology to constrain carbon dioxide emissions from coal-fired power plants using satellite observations of co-emitted nitrogen dioxide"

_Atmospheric Chemistry and Physics, 2019_

## Referee Comment (RC1) · Anonymous Referee #2 · 5 Aug 2019

Liu et al. describe a method to estimate CO2 emissions from power plants using satellite observations of tropospheric NO2 columns. The method involves the estimation of NOx emissions using a top-down approach previously developed by the authors and estimation of CO2 emissions by applying a NOx/CO2 emission ratio derived from direct stack emission measurements of both gases. The topic of the manuscript is important and relevant in the context of the ongoing development of the global emission monitoring system intended to support the elaboration of climate control and mitigation strategies. Although the idea to use satellite NO2 measurements to constrain CO2

emissions from fossil fuel burning is not new, application of this approach to specifically power plant emissions is a step forward. Another new point of the study is the analysis of the relationship between NOx and CO2 emissions from different types of coal-fired power plants in the US. That said, I keep wondering whether and how the method proposed in this manuscript can be proven useful in any scientific or practical applications. The weak points of the manuscript and my suggestions to the authors are outlined in my comments below.

Major comment

I find that the manuscript lacks clear logic in presenting the ideas and results of the authors. Specifically, while the main focus in Section 2 ("Method") is given to the analysis of the CEMS stack measurements in the US in the period from 2005 to 2017, it is not explained and justified how the outcome of this analysis can be used for applications outside of the US. Such possible applications are illustrated in the manuscript (in Sect. 3.3) by the example of only one power plant (Matimba), for which the authors use the NOx/CO2 emission ratio estimated only for 2005 and even argue that this estimate (based on the US data) is not directly applicable to the Matimba plant. Furthermore, if the "regressed" estimates of the NOx/CO2 emission ratio are not directly applicable to power plants outside of the US, the application of these approximate estimates to the selected 8 power plants inside of the US (presumably to test the method) seems to be pointless, as the CEMS measurements provide accurate direct estimates of the NOx/CO2 emission ratio for any power plant in the US. As for the Matimba power plant, a reasonable alternative to using the CEMS measurements would be to get a corresponding estimate of the NOx/CO2 emission ratio from the ODIAC inventory. Therefore, in the present form, the discussion and evaluation of the method is very confusing and, to some extent, misleading. In this respect, I recommend that the authors illustrate the potential of their method and the usefulness of the analysis of the US CEMS data by considering a few more power plants outside of the US (e.g., in China), paying special attention to the accuracy of the estimates of the NOx/CO2 emission ratio

based on the US CEMS data versus the accuracy of corresponding estimates that can be obtained directly from available data of global and regional emission inventories.

Specific comments

p.2, l.16-18: I believe that the narrow swath of the OCO-2 sensor is not the main reason for the limitations of the novel and promising method proposed by Reuter et al. (2019). I suggest that the authors provide a more extensive and accurate discussion (not necessarily in Introduction) of the advantages and disadvantages of their approach with respect to that of Reuter et al. (2019).

p.2, l.37: I recommend that the authors avoid boasting about the "novel" method here and elsewhere. Actually, the only significant new point of their method is that it is focused on a particular source of CO2 emissions (as noted above). A very similar method to constrain CO2 emissions is described in previous papers (cited in this manuscript) focused on estimating fossil fuel burning CO2 emissions in China and in Europe. Certainly, there are differences concerning the ways to estimate the NOx emissions and NOx/CO2 emission ratio in the different studies, but these differences are mostly of technical nature. Furthermore, the method which was used to estimate NOx emissions in this study is identical to that presented by the same authors in their previous papers.

p.3, l.7-12: It would be useful to explain briefly why a special approximation procedure is needed to estimate a NOx/CO2 emission ratio while using the CMES data (i.e. why the NOx/CO2 emission ratio for any given power plant in the US could not be directly evaluated using the corresponding CMES measurements).

P3. l.21. It is quite unusual and inconvenient that the first figure ever mentioned in the manuscript is Figure 5 (instead of Figure 1). The order of the figures should be corrected.

p.3, l.29-32: The authors should explain the origin and significance of the value "1.32". Would their estimates be less accurate if they assumed that the NOx/NO2 ratio equals,

say, to 1.3? Further, do the authors imply that if one had a way to measure the NO/NO2 ratio around any power plant anywhere in the world with a spatial resolution of 13 km×24 km, then the measured NOx/NO2 ratio would be exactly 1.32? Wouldn't the NOx/NO2 ratio actually strongly vary from site to site and would depend on the ozone level (which is frequently not determined by local pollution sources) and the age of the plume? Doesn't the fact that the estimates of the NOx lifetime inferred from satellite measurements vary across the 8 power plants within almost a factor of 2 (according to Table 2) mean that OH (and therefore O3) levels are quite different in plumes from different power plants? Overall, I believe that the uncertainty associated with the estimation of the NOx/NO2 ratio should be carefully discussed and evaluated (perhaps, using a chemistry transport model). A brief and superficial discussion of this important point in Liu et al. (2016) is certainly insufficient.

Table S1: The authors provided some useful supplementary information for Sect. 2.1 in Table S1, but this table is not mentioned and discussed anywhere in the manuscript.

p.4, l.3-33: I suggest the authors provide an additional figure illustrating the NO2 plume from the Rockport power plant along with a corresponding Gaussian fit.

p.5, l.5: It would be helpful if the authors explained here what is the purpose of creating "continuous and consistent records of ratio_CEMS...". Are these records supposed to be helpful for estimating CO2 emissions inside of the US (although accurate estimates of the NOx/CO2 ration are already provided by CEMS for each power plant) or outside of the US (although the applicability of the CEMS data outside of the US is very questionable)?

Sect. 3.2: In my opinion, the uncertainties of the emission estimates inferred from the OMI measurements are well characterized by the standard deviations reported in Table 3. However, these "data-based" uncertainty estimates are not discussed in the manuscript. The present discussion of the uncertainties, however, looks very superficial. I suggest the authors provide a separate table (e.g., in the Supporting information)

reporting the uncertainties associated with each power plant and with each individual factor contributing to the total uncertainty. Also, I wonder how a reader acquainted with the basic knowledge of the mathematical statistics is supposed to interpret the values of the uncertainty reported in this section. Do these values represent the standard deviation (that is, the confidence interval corresponding to the 68.3 percentile)? If so, does the fact that the uncertainty estimates range from 62%–96% mean that there is a significant chance that a true value of the emissions can be below zero (assuming that the error distribution is Gaussian)? My suggestion is to consider reporting the so huge uncertainties in terms of the geometric standard deviation (thus assuming that the error distribution is log-normal).

p.7, l.19,20: If the authors believe that the NOx/CO2 emission ratio at Matimba is on the upper end of the US values, then perhaps they should have used a maximum value of the NOx/CO2 emission ratios among all of the US power plants without NOx emission control. Anyway, it is not clear how the standard deviation of ratio_regressed was evaluated? Is it the standard deviation of the slope of a linear fit or the standard deviation of the original NOx/CO2 emission ratios from the CMES data?

p.7, l.29-31: According to Reuter et al. (2019), the CO2 emission estimates for the Matiba power plant are available also from the ODIAC inventory. The authors could consider using the corresponding estimates for comparison.

Conclusions: This section looks unusually short for ACP. Furthermore, instead of providing a clear and logical summary of the major findings of the study, the authors preferred to speculate about possible future developments of their method. Accordingly, I believe this section needs to be re-written and significantly extended.

Figure 2: Do the emissions shown in this figure correspond to the ozone season only? If so, this should be indicated in the figure caption. The regression coefficients could be reported only with one or two digits after the point. Is there a reason for showing a linear regression with the intercept term in the panel (c) and without the intercept in

other panels?

Figure 8: The meaning of a shaded band should be clearly explained in the figure caption. I suggest also to supply the emission estimates inferred from the OMI observations with the error bars corresponding to the mean of the standard deviations reported in Table 3.

---

## Referee Comment (RC2) · Anonymous Referee #1 · 6 Aug 2019

General comments

The manuscript presents a methodology to derive CO2 emissions using satellite-based NO2 retrievals from OMI instrument. The topic is very interesting as not many studies have successfully attempted space-based CO2 emission estimation (while much more common is the top-down emission estimation for short-lived gases such as NO2) and most of the previous studies only derive emissions for a few sites in the world. The results could be a good addition to the existing literature on the subject but I feel this work still does not dramatically improve what was achieved in previous studies in terms

of emission estimation from CO2 point sources. The methodology is reasonable but more effort should be put in proving how this approach could be extended to more than the 8 point sources analysed in the manuscript.

Therefore I would suggest to provide some sort of recommendations (or criteria) on how to apply the same approach to other point sources depending on the characteristics of the power plants. One possibility could be to test the approach on a few other cases outside US in addition to Matimba in order to illustrate the potential differences.

The manuscript can be published after addressing this issue and the following comments.

Specific comments

1. P2 L33 -> There is a recent update to this paper where the anomalies are calculated on global scale and also TROPOMI data are used for comparison on local scale. You might want to add this as well in your intro: Hakkarainen, J.; Ialongo, I.; Maksyutov, S.; Crisp, D. Analysis of Four Years of Global XCO2 Anomalies as Seen by Orbiting Carbon Observatory-2. Remote Sens. 2019, 11, 850.

Here also another work it might be worth mentioning: Wang, S., Zhang, Y., Hakkarainen, J., Ju, W., Liu, Y., Jiang, F., & He, W. ( 2018). Distinguishing anthropogenic CO2 emissions from different energy intensive industrial sources using OCO‐2 observations: A case study in northern China. Journal of Geophysical Research: Atmospheres, 123, 9462– 9473. https://doi.org/10.1029/2018JD029005

2. P7 L19-20 "We assume the NOx to CO2 emission ratio of Matimba is on the upper end of the US values, considering that it is not equipped with any NOx control devices, even low-NOx burners which are widely installed in US power plants" This step is quite critical if you think about extending the method to other sources. You are basically saying that you have to know already something on the source before applying the method. . . how do you expect to make this choice for other sources? Please comment.

3. Fig. 8 How do your emission estimates for Matimba compare with Reuter 2019 estimate?

4. P11 L25 This paper in now published: Reuter, M., Buchwitz, M., Schneising, O., Krautwurst, S., O'Dell, C. W., Richter, A., Bovensmann, H., and Burrows, J. P.: Towards monitoring localized $CO_2$ emissions from space: co-located regional $CO_2$ and $NO_2$ enhancements observed by the OCO-2 and S5P satellites, Atmos. Chem. Phys., 19, 9371-9383, https://doi.org/10.5194/acp-19-9371-2019, 2019.

5. Sect. 2.2 Is there any other dataset in addition to EPA's CEMS you could verify these ratio with?

Technical comments

P3 L20 "...plants.As discussed" there is a space missing here

P7 L11 I would change the title with "Application to Matimba power plant" or something like that more specific

---

## Author Comment (AC2) · 29 Oct 2019

The comment was uploaded in the form of a supplement:
https://www.atmos-chem-phys-discuss.net/acp-2019-521/acp-2019-521-AC2-supplement.pdf

---

## Author Response (AR1)

**Cover Letter**

Dear editor,

We have carefully addressed the thoughtful comments of the reviewers and believe that our manuscript is now much improved and ready for publication in ACP. In our paper, we present a methodology to infer $CO_2$ emissions from power plants using satellite observations of co-emitted $NO_2$. Reliable estimates of emissions of this important climate gas are necessary for predicting climate change and developing effective mitigation strategies.

Our work is timely as 1) it is currently not feasible to infer $CO_2$ emissions directly from satellite observations of $CO_2$ with current sensors, and 2) $CO_2$ emissions are not reliably measured (at stack) and reported for most power plants around the world. We demonstrated our methodology on eight US power plants. Though we were limited by current (OMI) sensor capabilities, we fully expect that our methodology will be more broadly applied to global power plants using improved $NO_2$ data from new and upcoming sensors (TROPOMI, TEMPO), which have improved signal-to-noise, finer spatial resolutions, etc.

I am looking forward to hearing from you.

Best regards,
Fei

*Anonymous Referee #1*
*General comments*
*The manuscript presents a methodology to derive $CO_2$ emissions using satellite-based $NO_2$ retrievals from OMI instrument. The topic is very interesting as not many studies have successfully attempted space-based $CO_2$ emission estimation (while much more common is the top-down emission estimation for short-lived gases such as $NO_2$) and most of the previous studies only derive emissions for a few sites in the world. The results could be a good addition to the existing literature on the subject but I feel this work still does not dramatically improve what was achieved in previous studies in terms of emission estimation from $CO_2$ point sources. The methodology is reasonable but more effort should be put in proving how this approach could be extended to more than the 8 point sources analysed in the manuscript.*
*Therefore I would suggest to provide some sort of recommendations (or criteria) on how to apply the same approach to other point sources depending on the characteristics of the power plants. One possibility could be to test the approach on a few other cases outside US in addition to Matimba in order to illustrate the potential differences.*
*The manuscript can be published after addressing this issue and the following comments.*

**Response:** We thank Referee #1 for the thoughtful comments. We have added Figure 8 to compare the US ratios derived in this study with the ratios for other countries from a bottom-up emission database. We have added Figure 9 to further clarify how to apply our approach to power plants outside the US. We have added a new subsection 3.3.1 to discuss the addition.

[Figure]

**Figure 8** Comparison of the regressed $NO_x$ to $CO_2$ emission ratios derived from the global power emissions database (GPED) for different regions versus the correlation coefficient of the regression. The blue and red circles denote regions that are subject to more strict standard for $NO_x$ emissions from power plants (i.e., a $NO_x$ ELV of 200 mg/m$^3$ or less) and other regions, respectively. Y axis: the slope of the regression of the $NO_x$ to $CO_2$ emissions with an assumed y-intercept of zero. Error bars show the standard deviations for the $NO_x$ to $CO_2$ emission ratios for individual power plants. X axis: correlation coefficient of the regression. The dashed line represents 2005 US $ratio_{regressed}^{CEMS}$ for bituminous coal derived in this study. The grey shadow represents 2005 US $ratio_{regressed}^{CEMS}$ $\pm$ standard deviation.

*China switched from being a less strict country to a more strict country in 2014, when most coal-fired power plants in China were required to comply with its new emission standards (GB13223-2011).

[Figure]

**Figure 9** Schematic of our methodology to estimate the $NO_x$ to $CO_2$ emission ratios for power plants outside the US.

*China switched from being a less strict country to a more strict country in 2014, when most coal-fired power plants in China were required to comply with its new emission standards (GB13223-2011).

To better place the importance of our work into context, we added the following paragraph to the conclusions:

"We found that it is feasible to infer $CO_2$ emissions from satellite $NO_2$ observations, but limitations of the current satellite data (e.g., spatio-temporal resolution, signal-to-noise) only allow us to apply our method to eight large and isolated U.S. power plants. Looking forward, we anticipate that these limitations will diminish for the recently launched (October 2017) European Union Copernicus Sentinel 5 precursor TROPOMI, and three upcoming (launches expected in the early 2020s) geostationary instruments (NASA TEMPO; European Space Agency and Copernicus Programme Sentinel-4; Korea Meteorological Administration Geostationary Environment Monitoring Spectrometer, GEMS), which are designed to have superior capabilities to OMI. For example, higher spatial and temporal resolutions will likely improve the estimation

of $NO_x$ emissions as well as allow for the separation of more power plant plumes from nearby sources, thus increasing the number of power plants available for analysis. Therefore, future work will be to apply our method to these new datasets, especially after several years of vetted data become available. Additional future work will include applying our ratio-regression method to other regions of the world with reliable CEMS information, such as Europe, Canada and, more recently, China, to develop a more reliable and complete database with region-specific ratios."

*Specific comments*
*1. P2 L33 -> There is a recent update to this paper where the anomalies are calculated on global scale and also TROPOMI data are used for comparison on local scale. You might want to add this as well in your intro: Hakkarainen, J.; Ialongo, I.; Maksyutov, S.; Crisp, D. Analysis of Four Years of Global XCO2 Anomalies as Seen by Orbiting Carbon Observatory-2. Remote Sens. 2019, 11, 850.*
*Here also another work it might be worth mentioning: Wang, S., Zhang, Y., Hakkarainen, J., Ju, W., Liu, Y., Jiang, F., & He, W. (2018). Distinguishing anthropogenic CO₂ emissions from different energy intensive industrial sources using OCO-2 observations: A case study in northern China. Journal of Geophysical Research: Atmospheres, 123, 9462–9473.*
*https://doi.org/10.1029/2018JD029005*

**Response:** We thank for the comments. We have added both references in the introduction of the revised manuscript.

*2. P7 L19-20 "We assume the $NO_x$ to $CO_2$ emission ratio of Matimba is on the upper end of the US values, considering that it is not equipped with any $NO_x$ control devices, even low-$NO_x$ burners which are widely installed in US power plants" This step is quite critical if you think about extending the method to other sources. You are basically saying that you have to know already something on the source before applying the method. . . how do you expect to make this choice for other sources? Please comment.*

**Response:** The aim of the method developed by this study is to simplify the information needed to derive the $NO_x$ to $CO_2$ ratio. The basic information needed for the method is generally available. We have clarified this in Section 3.3.1, as follows:

"The application of the method contributes to simplifying the information needed to derive a reasonable $NO_x$ to $CO_2$ emission ratio. In a bottom-up approach, we often need many details including coal type, coal quality, boiler firing type, $NO_x$ emission control device type, and operating condition of boiler and emission control device when calculating $NO_x$ and $CO_2$ emissions. As demonstrated in Section 2.2, the method developed in this study can derive a reasonable estimate of the ratio for power plants without post-combustion $NO_x$ control device by merely given coal type. Even for regions without reliable emission information, the information on coal type, particularly for large power plants, are very likely publicly available. For power plants installing post-combustion $NO_x$ control technology, we additionally require the removal efficiency of the device to derive the ratio. The removal efficiency of post-combustion $NO_x$ control devices is usually directly reported, as the operation of such devices is very expensive and is expected to be subject to strict quality control and assurance standards. "

*3. Fig. 8 How do your emission estimates for Matimba compare with Reuter 2019 estimate?*

**Response:** Our estimate for Matimba (including the nearby Medupi which has operated since 2015) is comparable to Reuter 2019 estimate. Our estimate is 1.9–3.0 Gg/h for 2016* (i.e., the period from 2015 to 2017); Reuter 2019 estimate is 3.5±0.8 Gg/h for 2018. It should be noted that the Medupi power plant started operation in 2015 with limited capacity and that it still has not reached its nominal capacity. Therefore, it is no surprise that our estimate is lower than Reuter 2019 estimate. We have added Reuter 2019 estimate in Figure 11 and the related discussion in Section 3.3.2, as follows:

"Figure 11 shows $E_{CO_2}^{Sat}$ derived in this study and other independent estimates reported in the literature, including two top-down (Nassar et al., 2017; Reuter et al., 2019) and three bottom-up estimates (Wheeler and Ummel, 2008; Tong et al., 2018; Oda et al., 2018). Despite the uncertainties associated with each of these methods, the $CO_2$ emissions estimates agree reasonably well."

[Figure]

**Figure 11** Comparison of $\mathbf{E_{CO_2}^{Sat}}$ (Gg/h) derived in this study with existing estimates for the Matimba power plant during 2005 to 2017. $\mathbf{E_{CO_2}^{Sat}}$ is inferred based on the $NO_x$ to $CO_2$ emissions ratio ranging from $\boldsymbol{ratio_{regressed}^{CEMS}}$ to $\boldsymbol{ratio_{regressed}^{CEMS}}$ + standard deviation of ratio. The upper and lower grey bands denote the emissions inferred from $\boldsymbol{ratio_{regressed}^{CEMS}}$ and $\boldsymbol{ratio_{regressed}^{CEMS}}$+ standard deviation of ratio, respectively. The grey dots and error bars show the mean of the upper and lower grey bands and their uncertainties, respectively.
[a]Emissions are estimated for 2009 by Wheeler and Ummel (2008); for 2010 by Tong et al. (2018); for 2014 and 2016 by Nassar at al. (2017); for 2016 by Reuter et al. (2019); and for 2012 and 2016 by Oda at al. (2018).

*4. P11 L25 This paper in now published: Reuter, M., Buchwitz, M., Schneising, O., Krautwurst, S., O'Dell, C. W., Richter, A., Bovensmann, H., and Burrows, J. P.: Towards monitoring localized CO2 emissions from space: co-located regional $CO_2$ and $NO_2$ enhancements observed by the OCO-2 and S5P satellites, Atmos. Chem. Phys., 19, 9371-9383, https://doi.org/10.5194/acp-19-9371-2019, 2019.*

**Response:** Thanks for pointing out this. We have updated the reference in the revised manuscript.

*5. Sect. 2.2 Is there any other dataset in addition to EPA's CEMS you could verify these ratio with?*

**Response:** EPA's CEMS has been widely used to develop emission inventories. To the best of our knowledge, all the widely-used regional and global bottom-up emission inventories adopt EPA's CEMS to estimate $NO_x$ and $CO_2$ emissions for US power plants. To the best of our knowledge, there is no independent dataset available to verify the derived ratios for the US power plants.

Technical comments
*P3 L20 ". . .plants.As discussed" there is a space missing here*

**Response:** Thanks for pointing out this. We have updated it in the revised manuscript.

P7 L11 I would change the title with "Application to Matimba power plant" or something like that more specific

**Response:** Thanks. We have changed the title for section 3.3.2 accordingly.

*Anonymous Referee #2*
*Liu et al. describe a method to estimate $CO_2$ emissions from power plants using satellite observations of tropospheric $NO_2$ columns. The method involves the estimation of $NO_x$ emissions using a top-down approach previously developed by the authors and estimation of $CO_2$ emissions by applying a $NO_x/CO_2$ emission ratio derived from direct stack emission measurements of both gases. The topic of the manuscript is important and relevant in the context of the ongoing development of the global emission monitoring system intended to support the elaboration of climate control and mitigation strategies. Although the idea to use satellite $NO_2$ measurements to constrain $CO_2$ emissions from fossil fuel burning is not new, application of this approach to specifically power plant emissions is a step forward. Another new point of the study is the analysis of the relationship between $NO_x$ and $CO_2$ emissions from different types of coal-fired power plants in the US. That said, I keep wondering whether and how the method proposed in this manuscript can be proven useful in any scientific or practical applications. The weak points of the manuscript and my suggestions to the authors are outlined in my comments below.*

**Response:** We thank Referee #2 for the thoughtful comments, which we address carefully below.

Major comment
*I find that the manuscript lacks clear logic in presenting the ideas and results of the authors. Specifically, while the main focus in Section 2 ("Method") is given to the analysis of the CEMS stack measurements in the US in the period from 2005 to 2017, it is not explained and justified how the outcome of this analysis can be used for applications outside of the US. Such possible applications are illustrated in the manuscript (in Sect. 3.3) by the example of only one power plant (Matimba), for which the authors use the $NO_x/CO_2$ emission ratio estimated only for 2005 and even argue that this estimate (based on the US data) is not directly applicable to the Matimba plant. Furthermore, if the "regressed" estimates of the $NO_x/CO_2$ emission ratio are not directly applicable to power plants outside of the US, the application of these approximate estimates to the selected 8 power plants inside of the US (presumably to test the method) seems to be pointless, as the CEMS measurements provide accurate direct estimates of the $NO_x/CO_2$ emission ratio for any power plant in the US. As for the Matimba power plant, a reasonable alternative to using the CEMS measurements would be to get a corresponding estimate of the $NO_x/CO_2$ emission ratio from the ODIAC inventory. Therefore, in the present form, the discussion and evaluation of the method is very confusing and, to some extent, misleading. In this respect, I recommend that the authors illustrate the potential of their method and the usefulness of the analysis of the US CEMS data by considering a few more power plants outside of the US (e.g., in China), paying special attention to the accuracy of the estimates of the $NO_x/CO_2$ emission ratio based on the US CEMS data versus the accuracy of corresponding estimates that can be obtained directly from available data of global and regional emission inventories.*

**Response:** We address the major comment as below.
- The significance of the method validation for US power plants:

In our study, we investigate the feasibility of using satellite data of $NO_2$ to infer $CO_2$ emissions, which could serve as a complementary verification of bottom-up inventories or be used to supplement these inventories.

We first apply our methodology to U.S. power plants, which have accurate CEMS emissions. We systematically identify sources of variation (i.e., coal type and type of $NO_x$ control device). The

high degree of accuracy of the U.S. CEMS data allows us to verify whether our methodology is feasible or not. In short, we found that it is feasible, but limitations of the current satellite data (e.g., spatio-temporal resolution, signal-to-noise) only allow us to apply our methodology to eight power plants.

Looking forward, we anticipate that current (e.g., TROPOMI) and future sensors (e.g., TEMPO, Sentinel-4, GEMS) will reduce the limitations of the satellite data, especially after their time records have lengthened, allowing us to apply our methodology to more the US and world power plants.

We have clarified this in the revised abstract, introduction and conclusion.

- The potential application of the method and the US ratio derived in this study:

CEMS measurements are available for some power plants in the US, Canada, European Union (EU) member states, Japan, South Korea, and, more recently, China. However, there is still a significant number of power plants in those countries without CEMS technology, particularly for $CO_2$ measurements. For example, EU member states do not require power plants to use CEMS for $CO_2$ reporting and the majority of plants in the EU therefore reports $CO_2$ emissions based on emission factors (Sloss, 2011). Therefore, we recommend applying our method described in Section 2.2 to infer region-specific ratios for those power plants. The method developed in this study provides a simplified but reliable method to determine the ratios for those power plants.

Many or most power plants in South America, Africa, and Asia (minus China) do not report CEMS measurements at all or their observations are of questionable quality. Therefore, bottom-up emission inventories for $NO_x$ and $CO_2$ from these countries are highly uncertain, confounding national and international efforts to design effective climate mitigation strategies. We have added a new subsection 3.3.1 to discuss how to apply the ratios derived in this study to other regions.

As suggested, we added the comparison of the ratios derived in this study with those in the global coal-fired power plant emissions database (GPED) in Section 3.3.1. GPED is the only publicly available bottom-up emission database reporting both $NO_x$ and $CO_2$ emissions for individual power plants all over the world. The US values show reasonable agreement with other countries' values identified by GPED. The details are as follows:

[revised manuscript text omitted]

- The recommendation of using the ODIAC inventory to derive the ratios.

We agree that ODIAC is a great source for $CO_2$ emissions. However, it does not provide $NO_x$ emissions. It is not practical to calculate the ratios based on ODIAC.

*Specific comments*
*p.2, l.16-18: I believe that the narrow swath of the OCO-2 sensor is not the main reason for the limitations of the novel and promising method proposed by Reuter et al. (2019). I suggest that the authors provide a more extensive and accurate discussion (not necessarily in Introduction) of the advantages and disadvantages of their approach with respect to that of Reuter et al. (2019).*

**Response:** We have added the discussion in the revised introduction, as follows:
"More recently, the co-located regional enhancements of $CO_2$ observed by OCO-2 and $NO_2$ observed by TROPOMI were analysed to infer localized $CO_2$ emissions for six hotspots including one power plant globally (Reuter et al., 2019). As emissions plumes are significantly longer than the swath width of OCO-2 (10km), OCO-2 sees only cross sections of plumes, which may not be sufficient to infer emission strengths. Because power plant emissions can have substantial temporal variations (Velazco et al., 2011) and the cross-sectional $CO_2$ fluxes are valid only for OCO-2 overpass times, the cross-sectional fluxes may not adequately represent the annual or monthly averages, which are required for the development of climate mitigation strategies. In addition, the cross-sectional fluxes may not be a good approximation for emission strengths if meteorological conditions are not taken into account (Varon et al., 2018). As compared to the method proposed in this study, Reuter's method has the advantage of not requiring a priori emission information. However, there are currently no satellite instruments with a wide enough swath to allow wider application of Reuter's method. "

*p.2, l.37: I recommend that the authors avoid boasting about the "novel" method here and elsewhere. Actually, the only significant new point of their method is that it is focused on a*

*particular source of $CO_2$ emissions (as noted above). A very similar method to constrain $CO_2$ emissions is described in previous papers (cited in this manuscript) focused on estimating fossil fuel burning $CO_2$ emissions in China and in Europe. Certainly, there are differences concerning the ways to estimate the $NO_x$ emissions and $NO_x/CO_2$ emission ratio in the different studies, but these differences are mostly of technical nature. Furthermore, the method which was used to estimate $NO_x$ emissions in this study is identical to that presented by the same authors in their previous papers.*

**Response:** We have deleted the term of novel in the revised manuscript.

*p.3, l.7-12: It would be useful to explain briefly why a special approximation procedure is needed to estimate a $NO_x/CO_2$ emission ratio while using the CMES data (i.e. why the $NO_x/CO_2$ emission ratio for any given power plant in the US could not be directly evaluated using the corresponding CMES measurements).*

**Response:** CEMS measurements are available for some power plants in the US, Europe, Canada and, more recently, China. For those power plants with CEMS measurements, we agree that it is more straightforward and accurate to use the measured values. However, there is still a significant number of power plants in those countries without CEMS technology, particularly for $CO_2$ measurements. The method developed by this study provides a more reliable method to determine the ratios for those power plants without CEMS based on CEMS data for other plants. We have clarified this in the revised Section 3.3.1, as follows:

"More strict countries, including Canada, European Union (EU) member states, Japan, South Korea, and, more recently, China, usually use CEMS to monitor emissions, particularly from the largest emitters. For power plants with CEMS measurements for both $NO_x$ and $CO_2$ emissions, it is straightforward to use the measured ratios. However, there is still a significant number of power plants in those countries without CEMS technology, particularly for $CO_2$ measurements. For example, EU member states do not require power plants to use CEMS for $CO_2$ reporting and the majority of plants in the EU therefore reports $CO_2$ emissions based on emission factors (Sloss, 2011). Therefore, we recommend applying our method described in Section 2.2 to infer region-specific ratios for those power plants. The US $ratio_{regressed}^{CEMS}$ could be a less accurate, but reasonable approximation when no CEMS data are available, considering those countries share similar $NO_x$ ELVs for power plants as the US."

*P3. l.21. It is quite unusual and inconvenient that the first figure ever mentioned in the manuscript is Figure 5 (instead of Figure 1). The order of the figures should be corrected.*

**Response:** Thanks. We have reordered the figures in the revised manuscript.

*p.3, l.29-32: The authors should explain the origin and significance of the value "1.32". Would their estimates be less accurate if they assumed that the $NO_x/NO_2$ ratio equals, say, to 1.3? Further, do the authors imply that if one had a way to measure the $NO/NO_2$ ratio around any power plant anywhere in the world with a spatial resolution of 13 km×24 km, then the measured $NO_x/NO_2$ ratio would be exactly 1.32? Wouldn't the $NO_x/NO_2$ ratio actually strongly vary from*

*site to site and would depend on the ozone level (which is frequently not determined by local pollution sources) and the age of the plume? Doesn't the fact that the estimates of the $NO_x$ lifetime inferred from satellite measurements vary across the 8 power plants within almost a factor of 2 (according to Table 2) mean that OH (and therefore $O_3$) levels are quite different in plumes from different power plants? Overall, I believe that the uncertainty associated with the estimation of the $NO_x/NO_2$ ratio should be carefully discussed and evaluated (perhaps, using a chemistry transport model). A brief and superficial discussion of this important point in Liu et al. (2016) is certainly insufficient.*

**Response:** The number of 1.32 used for scaling up the $NO_2$ to $NO_x$ is based on the typical assumptions made in the section 6.5.1 of Seinfeld and Pandis (2006) for "typical urban conditions and noontime sun" following the recommendation by Beirle et al. (2011). We agree that the $NO/NO_2$ ratio might vary locally. But these local variations are not expected to be significant over spatial scales of ~100−200 km and annual temporal averaging. We included increased uncertainty of the $NO_x/NO_2$ ratio from 10% to 20% when calculating the overall uncertainties. We recognize that uncertainties resulting from the $NO_x/NO_2$ ratio may be better understood when more direct measurements are available in the future. We have clarified this in the Section 3.2 of the revised manuscript, as follows:

"The number of 1.32 used for scaling the $NO_2$ to $NO_x$ ratio is based on assumptions presented in section 6.5.1 of Seinfeld and Pandis (2006) for "typical urban conditions and noontime sun". Note that conditions are quite similar in this study because of the overpass time of OMI close to noon, the selection of cloud-free observations, the focus on the ozone season, and the focus on polluted regions. A case study of CTM simulations shows an identical value of 1.32 for Paris in summer (Shaiganfar et al., 2017). The simulated $NO_x/NO_2$ ratio at the OMI overpass time within the boundary layer (up to 2 km) in a chemistry–climate model, EMAC (Jöckel et al., 2016), was $1.28 \pm 0.08$ for polluted ($NO_x > 1 \times 10^{15}$ molec cm$^{-2}$) regions for the July 1, 2005, and $1.32 \pm 0.06$ on average for the ozone season. However, the coarse grid of EMAC ($2.8° \times 2.8°$ in latitude and longitude) may not capture the true range of variation of the $NO_x/NO_2$ ratio. Therefore, we assumed an uncertainty of 20% arising from the $NO_x/NO_2$ ratio, double than the standard deviation of the EMAC ratio. "

*Table S1: The authors provided some useful supplementary information for Sect. 2.1 in Table S1, but this table is not mentioned and discussed anywhere in the manuscript.*

**Response:** Thanks for pointing out this. We have introduced the table in the revised manuscript, as follows:
"The locations of the 8 plants are shown in Figure 1 and given in Table S1."
"The fitted lifetimes and other fitting parameters for all power plants are given in Table S1."

*p.4, l.3-33: I suggest the authors provide an additional figure illustrating the $NO_2$ plume from the Rockport power plant along with a corresponding Gaussian fit.*

**Response:** Thanks. We have added it as Figure 2 in the revised manuscript.

[Figure]

**Figure 2** Mean OMI NO$_2$ tropospheric VCDs around the Rockport power plant (Indiana, USA) for (a) calm conditions, (b) northeasterly winds and (c) their difference (northeasterly − calm) for the period of 2005 – 2017. The location of Rockport is labelled by a black dot. (d) NO$_2$ line densities around Rockport. Crosses: NO$_2$ line densities for calm (blue) and northeasterly winds (red) as function of the distance x to Rockport center. Grey line: the fit result. The numbers indicate the net mean wind velocities (windy − calm) from MERRA-2 (w) and the fitted lifetime τ.

*p.5, l.5: It would be helpful if the authors explained here what is the purpose of creating "continuous and consistent records of ratio_CEMS…". Are these records supposed to be helpful for estimating CO$_2$ emissions inside of the US (although accurate estimates of the NO$_x$/CO$_2$ ration are already provided by CEMS for each power plant) or outside of the US (although the applicability of the CEMS data outside of the US is very questionable)?*

**Response:** The sentence indicates that $ratio^{CEMS}$ for plants prior to and after installing post-combustion NO$_x$ control systems is continuous and consistent, because the estimation is based on

$ratio_{regressed}^{CEMS}$ for plants without post-combustion control systems in operation. We have deleted the terms of continuous and consistent in the revised manuscript to prevent misunderstanding.

*Sect. 3.2: In my opinion, the uncertainties of the emission estimates inferred from the OMI measurements are well characterized by the standard deviations reported in Table 3. However, these "data-based" uncertainty estimates are not discussed in the manuscript. The present discussion of the uncertainties, however, looks very superficial. I suggest the authors provide a separate table (e.g., in the Supporting information) reporting the uncertainties associated with each power plant and with each individual factor contributing to the total uncertainty. Also, I wonder how a reader acquainted with the basic knowledge of the mathematical statistics is supposed to interpret the values of the uncertainty reported in this section. Do these values represent the standard deviation (that is, the confidence interval corresponding to the 68.3 percentile)? If so, does the fact that the uncertainty estimates range from 62%–96% mean that there is a significant chance that a true value of the emissions can be below zero (assuming that the error distribution is Gaussian)? My suggestion is to consider reporting the so huge uncertainties in terms of the geometric standard deviation (thus assuming that the error distribution is log-normal).*

**Response:** We agree that the standard deviation reported in Table 3 is a good indicator of the uncertainty. We also calculate the geometric standard deviation of the difference between $E_{CO_2}^{CEMS}$ and $E_{CO_2}^{Sat}$ from 2006* to 2016* for individual power plants in Table S2 as an alternative measure to reflect the uncertainty following the suggestion of the reviewer. In the revised manuscript, we have added the discussion on this "data-based" uncertainty analysis, as follows:

"The mean and the standard deviation of the relative differences between $E_{NO_x}^{CEMS}$ and $E_{NO_x}^{Sat}$, and $E_{CO_2}^{CEMS}$ and $E_{CO_2}^{Sat}$ for all eight power plants provide a good alternative measure of uncertainties (Table 3). The relative differences are rather small, which are 0% $\pm33\%$ and 8% $\pm41\%$ (mean $\pm$ standard deviation) for $NO_x$ and $CO_2$, respectively. We additionally calculate the geometric standard deviations (GSDs) of the difference between $E_{CO_2}^{CEMS}$ and $E_{CO_2}^{Sat}$ from 2006* to 2016* for individual power plants in Table S2. The small values of GSDs ranging from 1.07 to 1.31 further improve our confidence in the accuracy of the derived emissions in this study."

We have added a separate Table S2 to list the contributors to the overall uncertainties as suggested by the reviewer. We report the derived uncertainties as a 95% confidence interval (CI). Note that we adjust our uncertainty estimates for some contributors. We increased the uncertainty of the $NO_x/NO_2$ ratio from 10% to 20% (see response to the comments on the $NO_x/NO_2$ ratio). We decreased the uncertainty arising from the variations of fitted lifetimes by wind direction from 40% to 20%, because the average of the standard deviation of lifetimes for all wind directions decreased from 40% in Liu et al. (2016) to 20% in this study. The details are given in section 1 of the supplement, as follows:
"The uncertainty analysis is similar to the procedure described in our previous work (Liu et al., 2016), based on the fit performance and the dependencies on the a priori settings as determined in sensitivity studies. We report the derived uncertainties as a 95% confidence interval (CI). Here we briefly list the sources of uncertainties and how they are quantified. Further details are provided in Section 3 of the Supplement of Liu et al. (2016). In summary, we conclude that:

- *Choice of integration and fit intervals:* Uncertainties arising from the choice of integration and fit intervals are about 10% for the lifetime and 20% for the total $NO_2$ mass, respectively, based on our sensitivity analysis by changing integration and fit intervals.
- *Fit errors:* The fit errors expressed as 95% confidence interval (CI) are derived from the least-squares fit routine directly for individual sources. They are typically on the order of 10% for both lifetime and total $NO_2$ mass, both of which are propagated into the uncertainty of $E_{NO_x}^{Sat}$. In addition, the standard deviation of fitted lifetimes for all wind direction sectors is regarded as a measure of uncertainty to reflect the reliability of lifetimes, which is 20% on average for all power plants.
- *Wind fields:* The uncertainty associated with the wind data is 30%. The choice of wind layer height and the uncertainties of wind fields themselves contribute to the overall uncertainty.
- The derived $NO_x$ emissions are affected by the uncertainty of the $NO_2$ tropospheric VCDs (~30%) and the $NO_x/NO_2$ ratio (~20%).
- Effects of a possible systematic change of $NO_2$ tropospheric VCDs from calm to windy conditions result in an uncertainty of ~10%.
- $ratio_{regressed}^{CEMS}$ contributes to an uncertainty of 15%.
- For power plants with post-combustion $NO_x$ control devices, an additional uncertainty of 20% comes from the predicted $NO_x$ removal efficiency of the devices.

The uncertainties of each contributor for individual power plants are listed in Table S2. We assume that their contributions to the total uncertainty are independent and define the total uncertainty as the root of the quadratic sum of the aforementioned contributions."

*p.7, l.19,20: If the authors believe that the $NO_x/CO_2$ emission ratio at Matimba is on the upper end of the US values, then perhaps they should have used a maximum value of the $NO_x/CO_2$ emission ratios among all of the US power plants without $NO_x$ emission control. Anyway, it is not clear how the standard deviation of ratio_regressed was evaluated? Is it the standard deviation of the slope of a linear fit or the standard deviation of the original $NO_x/CO_2$ emission ratios from the CMES data?*

**Response:** We assume the $NO_x$ to $CO_2$ emission ratio of Matimba is on the upper end of the US values, considering South Africa has not implemented improvements in boiler operations to decrease the ratio, such as optimizing furnace design and operating conditions, as in the US. We thus use the ratio for year 2005, instead those for more recent years to infer $CO_2$ emissions for the entire period. The standard deviation is that of the $NO_x/CO_2$ emission ratios for individual power plants from CEMS. We believe the ratio of Matimba is more likely to range from 2005 $ratio_{regressed}^{CEMS}$ to 2005 $ratio_{regressed}^{CEMS}$ + standard deviation, instead of being 2005 $ratio_{regressed}^{CEMS}$, considering that it is not equipped with any $NO_x$ control devices, even low-$NO_x$ burners which are widely installed in US power plants. But we don't use the maximum value of the ratio in this study, because it may be related with some plant-specific operating conditions, which is not applicable to other plants. We have clarified how to apply the ratios derived in this study to other regions in Section 3.3.1.

*p.7, l.29-31: According to Reuter et al. (2019), the $CO_2$ emission estimates for the Matiba power plant are available also from the ODIAC inventory. The authors could consider using the corresponding estimates for comparison.*

**Response:** Thanks. We have added the estimates from ODIAC in Figure 11 of the revised manuscript. $E_{CO_2}^{Sat}$ derived in this study shows reasonable agreement with the ODIAC.

*Conclusions: This section looks unusually short for ACP. Furthermore, instead of providing a clear and logical summary of the major findings of the study, the authors preferred to speculate about possible future developments of their method. Accordingly, I believe this section needs to be re-written and significantly extended.*

**Response:** We have extended the conclusion substantially to provide a summary of the major findings, as follows:

"In our study, we investigated the feasibility of using satellite data of $NO_2$ from power plants to infer co-emitted $CO_2$ emissions, which could serve as complementary verification of bottom-up inventories or be used to supplement these inventories that are highly uncertain in many regions of the world. For example, our estimates will serve as an independent check of $CO_2$ emissions that will be inferred from satellite retrievals of future $CO_2$ sensors (Bovensmann et al., 2010). Currently, uncertainties in $CO_2$ emissions from power plants confound national and international efforts to design effective climate mitigation strategies.

We estimate $NO_2$ and $CO_2$ emissions during the "ozone season" from individual power plants from satellite observations of $NO_2$ and demonstrate its utility for US power plants, which have accurate CEMS with which to evaluate our method. We systematically identify the sources of variation, such as types of coal, boiler, and $NO_x$ emission control device, and change in operating conditions, which affect the $NO_x$ to $CO_2$ emissions ratio. Understanding the causes of these variations will allow for better informed assumptions when applying our method to power plants that have no or uncertain information on the factors that affect their emissions ratios. For example, we estimated $CO_2$ emissions from the large and isolated Matimba power plant in South Africa, finding that our emissions estimate shows reasonable agreement with other independent estimates.

We found that it is feasible to infer $CO_2$ emissions from satellite $NO_2$ observations, but limitations of the current satellite data (e.g., spatio-temporal resolution, signal-to-noise) only allow us to apply our method to eight large and isolated U.S. power plants. Looking forward, we anticipate that these limitations will diminish for the recently launched (October 2017) TROPOMI, and three upcoming (launches expected in the early 2020s) geostationary instruments (NASA TEMPO; European Space Agency and Copernicus Programme Sentinel-4; Korea Meteorological Administration Geostationary Environment Monitoring Spectrometer, GEMS), which are designed to have superior capabilities to OMI. As demonstrated in Ialongo et al. (2019), high resolution TROPOMI observations are capable of describing the spatio-temporal variability of $NO_2$, even in a relatively small city like Helsinki. Higher spatial and temporal resolutions will likely reduce uncertainties in estimates of $NO_x$ emissions as well as allow for the separation of more power plant plumes from nearby sources, thus increasing the number of power plants available for analysis. Therefore, future work will be to apply our method to these

new datasets, especially after several years of vetted data become available. Additional future work will include applying our method to other regions of the world with reliable CEMS information, such as Europe, Canada and, more recently, China, to develop a more reliable and complete database with region-specific ratios. "

*Figure 2: Do the emissions shown in this figure correspond to the ozone season only? If so, this should be indicated in the figure caption. The regression coefficients could be reported only with one or two digits after the point. Is there a reason for showing a linear regression with the intercept term in the panel (c) and without the intercept in other panels?*

**Response**: For comparison to $E_{NO_x}^{Sat}$ and $E_{CO_2}^{Sat}$, we use emissions averaged over the ozone season derived from Air Markets Program Data (available at https://ampd.epa.gov/ampd/). However, Air Markets Program Data do not provide information about each plant's boiler firing types (e.g., tangential or wall-fired boiler), $NO_x$ control device type, fossil fuel type (with categories of coal, oil, gas and other), and coal type (with categories of bituminous, lignite, subbituminous, refined and waste coal), which are required to get reasonable ratio. Thus, we choose eGRID as the data source for Figure2. We use eGRID annual emissions in Figure 2, because eGRID does not provide $CO_2$ emissions specifically for the ozone season.

We have changed the regression coefficients to two digits after the point. We intent to show the linear regression without intercept in panels (a) and (b), because the regression slope was calculated requiring zero intercept for deriving $\boldsymbol{ratio_{regressed}^{CEMS}}$.

*Figure 8: The meaning of a shaded band should be clearly explained in the figure caption. I suggest also to supply the emission estimates inferred from the OMI observations with the error bars corresponding to the mean of the standard deviations reported in Table 3.*

**Response:** We have added the explanation for the shaded band in the revised caption, as follows: "
[revised manuscript text omitted]

| 2006* | 15% | 29% | 17% | 39% |
| 2007* | 10% | 29% | 16% | 38% |
| 2008* | 5%  | 30% | 14% | 39% |
| 2009* | -3% | 34% | 6%  | 39% |
| 2010* | -1% | 38% | 9%  | 46% |
| 2011* | -5% | 31% | 3%  | 40% |
| 2012* | -3% | 31% | 5%  | 41% |
| 2013* | -4% | 38% | 4%  | 49% |
| 2014* | -3% | 36% | 7%  | 46% |
| 2015* | -8% | 35% | 2%  | 41% |
| 2016* | -2% | 29% | 8%  | 22% |